# SmartFRZ: An Efficient Training Framework using Attention-Based Layer Freezing

**Sheng Li**[*1], **Geng Yuan**[*2], **Yue Dai**[*1], **Youtao Zhang**[1], **Yanzhi Wang**[2], **Xulong Tang**[1]
[1]University of Pittsburgh    [2]Northeastern University
{shl188,yud42,xulongtang}@pitt.edu
{yuan.geng,yanz.wang}@northeastern.edu    zhangyt@cs.pitt.edu

## Abstract

There has been a proliferation of artificial intelligence applications, where model training is key to promising high-quality services for these applications. However, the model training process is both time-intensive and energy-intensive, inevitably affecting the user's demand for application efficiency. Layer freezing, an efficient model training technique, has been proposed to improve training efficiency. Although existing layer freezing methods demonstrate the great potential to reduce model training costs, they still remain shortcomings such as lacking generalizability and compromised accuracy. For instance, existing layer freezing methods either require the freeze configurations to be manually defined before training, which does not apply to different networks, or use heuristic freezing criteria that is hard to guarantee decent accuracy in different scenarios. Therefore, there lacks a generic and smart layer freezing method that can automatically perform "in-situation" layer freezing for different networks during training processes. To this end, we propose a generic and efficient training framework (SmartFRZ). The core proposed technique in SmartFRZ is attention-guided layer freezing, which can automatically select the appropriate layers to freeze without compromising accuracy. Experimental results show that SmartFRZ effectively reduces the amount of computation in training and achieves significant training acceleration, and outperforms the state-of-the-art layer freezing approaches.

## 1 Introduction

Deep neural networks (DNNs) have become the core enabler of a wide spectrum of AI applications, such as natural language processing (Vaswani et al., 2017; Kenton & Toutanova, 2019), visual recognition (Li et al., 2022; Faraki et al., 2021), automatic machine translation (Sun et al., 2020; Zheng et al., 2021), and also the emerging application domains such as robot-assisted eldercare (Do et al., 2018; Bemelmans et al., 2012), mobile diagnosis (Panindre et al., 2021; Abdel-Basset et al., 2020), and wild surveillance (Akbari et al., 2021; Ke et al., 2020). To satisfy the growing demand for adaptability and training efficiency of DNN models in deployment, a surge of research efforts has been devoted to designing efficient training paradigms (He et al., 2021; Yuan et al., 2021; Wu et al., 2020; 2021). For example, sparse training (Evci et al., 2020; Yuan et al., 2021) and low-precision training (Yang et al., 2020; Zhao et al., 2021) are two active research areas for efficient training that can effectively reduce training costs, such as computing FLOPs and memory. However, there are still fundamental limitations in the generality of these methods when used in practice. Concretely, sparse training methods generally require the support of dedicated libraries or compiler optimizations (Chen et al., 2018; Niu et al., 2020) to leverage the model sparsity and save computation costs. Similarly, low-precision training (e.g., INT-8 or fewer bits) is hard to be supported by GPUs and requires specialized designs for edge devices such as field-programmable gate arrays (FPGAs) and application-specific integrated circuits (ASICs). Therefore, it is desirable to have a generic efficient training method that can be easily and effectively adapted to various application scenarios.

Recent studies (Brock et al., 2017; Liu et al., 2021) revealed that freezing some DNN layers at a certain stage (i.e., training iteration) during the training process will not degrade the accuracy of

---

[*]These authors contributed equally.

the final trained model and can effectively reduce training costs. Most importantly, layer freezing can be achieved using native training frameworks such as PyTorch/TensorFlow without additional support, making it more accessible to wide applications and users for training costs reduction and acceleration. Previous work has mainly used heuristic freezing strategies, such as empirically selecting which DNN layers to freeze and when to freeze (Lee et al., 2019; Yuan et al., 2022). As such, these heuristic freezing strategies require a trial-and-error process to find appropriate freezing configurations for individual tasks/networks, resulting in inconvenience and inefficiency when deployed to various application scenarios. Some recent works attempt to freeze the DNN layers in an adaptive manner by using the gradient-norm (Liu et al., 2021) or SVCCA score (He et al., 2021). However, it is known that DNN models generally do not monotonically converge to their optimal position. As a result, these adaptive methods, which decide whether the layers are frozen or not using the heuristic criteria, are less robust to the fluctuation of model training, leading to compromised accuracy. Therefore, we raise a fundamental question that has seldom been asked:

> *Is there a layer freezing method that can overcome the above-mentioned shortcomings while keeping the method efficient?*

Inspired by the outstanding performance of attention mechanism in solving sequence classification problems such as classification tasks (Long et al., 2018), dialog detection (Shen & Lee, 2016), and affect recognition (Gorrostieta et al., 2018), it is possible that the attention mechanism could also be a promising solution for layer freezing in efficient training, but it has never been explored in prior literature. In this paper, we innovatively introduce attention mechanism for layer freezing. In specific, we design a lightweight attention-based predictor to collect and rank the DNN context information from multiple timestamps during training process. Based on the prediction, we adaptively freeze DNN layers to save training computation costs and accelerate training process while maintaining high model accuracy. To train the attention-based predictor, we propose a layer representational similarity-based method to generate a special training dataset using publicly available dataset (e.g., ImageNet). Then, the predictor is trained offline once, and learns the generic convergence pattern along the training history, which can be generalized across different models and datasets. We summarize our contributions as:

- We design a novel and lightweight predictor using attention mechanism for layer freezing in efficient training. The predictor automatically captures DNN context information from multiple timestamps and adaptively freezes the layers during the training process.

- We propose to leverage the layer representational similarity to generate a special dataset for training the attention-based predictor. The trained predictor can be used for different datasets and networks.

- Combining the attention-based predictor design and its training method, we propose a generic efficient training framework, namely **SmartFRZ**, which can effectively reduce the training costs and accelerate training time. Specifically, for fine-tuning scenarios, Smart-FRZ consistently achieves higher training acceleration while maintaining similar or higher accuracy compared to the prior works. For training from scratch scenarios, our Smart-FRZ shows more significant advantages compared to the baseline methods including full training, linear freezing, and AutoFreeze, which achieves 0.16%, 2.65%, and 4.82% higher accuracy with 24%, 13%, and 15% less training time, respectively.

## 2 BACKGROUND AND RELATED WORK

**Layer Freezing.** Recent researches (Brock et al., 2017; Kornblith et al., 2019) have found that not all layers in deep nerual networks need to be trained equally. For example, the early layers in DNNs are responsible for low-level features extraction and usually have fewer parameters than the later layers, making the early layers converge faster during training. Therefore, the layer freezing techniques are proposed, which stop updating certain layers during the training process to save the training costs (Lee et al., 2019; Zhang & He, 2020).

The previous work mainly uses heuristic strategies to determine which layers to freeze and when to freeze them. For example, Brock et al. (2017) uses a linear/cubic schedule to freeze the layers sequentially (one by one from the first layers to the later layers). The recent work Yuan et al. (2022)

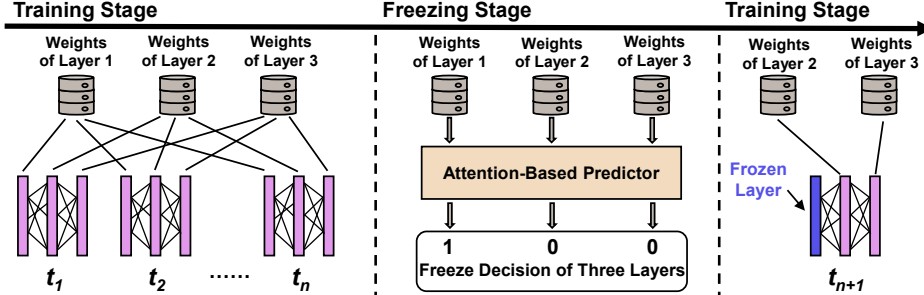

Figure 1: Overview of our attention-based layer freezing framework SmartFRZ. SmartFRZ continuously records the weights of all actively being-trained layers during the training process. Periodically, there will be a freezing stage, when SmartFRZ applies an attention-based predictor to determine whether each active layer can be frozen at this moment, based on the collected historical data. After the freezing stage, the model training process continues, and the frozen layer will not be updated and its weights will not be recorded anymore. In this figure, we assume that the attention-based predictor predicts to freeze layer 1 only.

also freezes layer by layer in a constant interval, but uses target training FLOPs reduction to decide the time to start freezing. Another branch of works attempt to freeze DNN layers adaptively and use the criteria such as gradient-norm (Liu et al., 2021) or SVCCA score (He et al., 2021) to control the freezing. However, all of these heuristic-based freezing strategies require manually designed freezing configurations or threshold, which can be varied between different networks and datasets. In this paper, we intend to address these shortcomings and propose a generic freezing method that can be adopted to different scenarios.

**Training Costs Reduction through Layer Freezing.** Layer freezing technique can save training costs including *computation FLOPs* and *memory accesses*. Generally, each iteration of DNN training consists of forward and backward propagation. On the one hand, once a layer is frozen, it still needs to compute the forward propagation since the subsequent layers still need its output features as their input features. Therefore, layer freezing cannot save training costs in forward propagation. On the other hand, the backward propagation consists of two parts of computations: i) calculating the gradients of weights and ii) calculating the gradients of activations. A frozen layer can always eliminate the computation cost and memory cost of the former part since the weights in the layer will not be updated anymore. However, the computation costs and memory costs of the latter part can only be eliminated if all the predecessor layers earlier than the current layer are frozen as well. This is because if a predecessor layer is not frozen, it needs to calculate the gradients of activations of the current layer to maintain the data pass through the backward propagation. Compared to the other efficient training techniques such as pruning or quantization, one significant advantage of layer freezing is that it does not require dedicated supports such as compiler optimizations or specialized hardware. It can be easily integrated into the general deep learning frameworks (e.g., PyTorch) and achieve almost linear acceleration according to the computation FLOPs reduction (Yuan et al., 2022). Meanwhile, it focuses on layers as granularity which is more efficient in reducing the training cost, and potentially, it can be combined with pruning and quantization to bring combined benefits.

## 3 SMARTFRZ FRAMEWORK DESIGN

### 3.1 FRAMEWORK OVERVIEW

In this section, we introduce our proposed layer freezing framework for efficient training, which can automatically freeze appropriate network layers to reduce unnecessary computation during the training process without compromising accuracy. Figure 1 shows the overview of the SmartFRZ framework. The core component of our framework is a lightweight attention-based predictor, which can automatically decide which layers will be frozen and when to freeze them during the training process. However, the attention-based predictor cannot be directly trained using conventional dataset for the classification tasks such as ImageNet. Therefore, we propose a novel method to generate the training dataset. We adopt offline training to train the predictor. Once the predictor is well-trained, it can be used for different datasets and networks since it learns generic converging patterns from collected training histories.

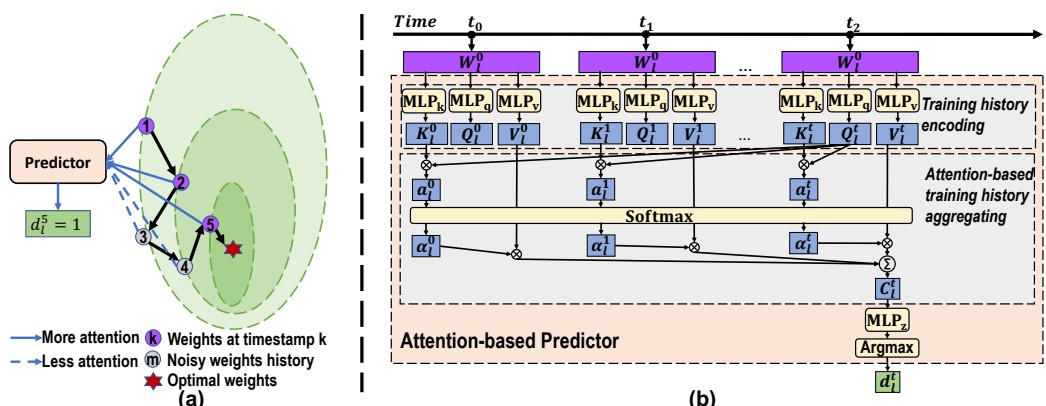

Figure 2: (a) An illustrated example of noisy weight history in which a darker shade within the circle indicates a lower loss. The training loss decreases at iteration $\{1, 2, 5\}$, yet increases at iterations $\{3, 4\}$. So the predictor needs to pay more attention to the weight history from iteration $\{1, 2, 5\}$ and less attention to the weight history from iterations $\{3, 4\}$ since the latter converge towards less optimal directions. (b) Detailed workflow of the attention-based predictor. At timestamp $t$, the predictor decides whether to freeze a specific layer $l$ in three steps. First, it encodes the training history independently into feature vectors (i.e., $K_l^j$, $Q_l^j$, $V_l^j$). Second, it computes attention scores $\alpha_l^j$ and aggregates the historical state feature into a context vector $C_l^t$. Third, it predicts the confidence scores of freezing or not and selects the decision with higher confidence.

In the following parts of this section, we will first introduce the design principle and workflow of our attention-based predictor in Section 3.2 and Section 3.3. Then, we will introduce how to create the special training dataset for the predictor in Section 3.4.

## 3.2 DESIGN PRINCIPLE OF THE ATTENTION-BASED PREDICTOR

The goal of the predictor is to decide whether to freeze a layer or not at a specific training iteration. While existing approaches measure the gradient norm change between two timestamps (Liu et al., 2021), the patterns of long-term converging history are missed. To this end, it is desirable to leverage the long-term weight history sequence for the prediction. Specifically, the predictor is designed to capture the generic converging pattern within the model training histories. We can define the task of the predictor as follows:

**Freezing Prediction Task (Definition #1).** For a layer $l$, whose weights at timestamp $j$ can be denoted as $W_l^j$: Given a sequence of its weight history $(W_l^j)_{j=0}^t$ at timestamp $t$, yield positive decision to freeze the layer at the current iteration (i.e., $d_l^t = 1$) if the layer is ready to be frozen, and yield negative decision (i.e., $d_l^t = 0$) if the layer needs further training.

The task is non-trivial due to noisy weight history information within the input sequence. Since the training model does not monotonically converge to the optimal solution, there exist scenarios such that: the history weight $W_l^n$ from iteration $n$ is less optimal than the history weight $W_l^m$ from iteration $m$, even if the $W_l^n$ is updated from $W_l^m$ (i.e., $n > m$). We call weights like $W_l^n$ as noisy weight history. These noisy weight histories represent incorrect converging directions, thus introducing unnecessary noises. We illustrate an example of the scenario in Figure 2(a). To this end, the predictor should avoid leveraging these misleading weight histories and selectively focus on other weight histories that provide more accurate training information toward the optimal weights.

We apply an attention-based predictor to meet the requirement. The predictor ranks the information from each timestamp (i.e., sampled iteration) and adaptively aggregates weight history from the input sequence. The aggregation is input-dependent and can select the history from those preferred timestamps. The detailed workflow is discussed in Section 3.3.

**Layer Tailoring.** To minimize the overheads introduced by the predictor, we need to make the predictor lightweight. Instead of assigning individual predictors for each layer, we intend to use only one predictor to serve different layers, thanks to the attention mechanism that makes the predictor can learn the generic convergence pattern. However, this raises another issue: *how to fit weight information with different sizes from different layers into a fixed-sized MLP predictor?*

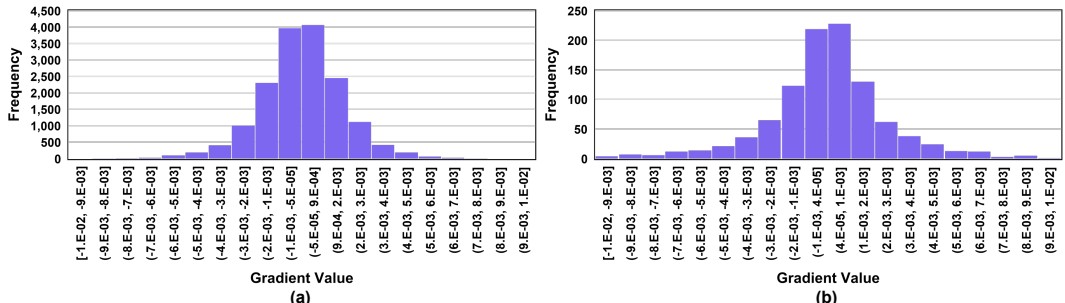

Figure 3: The histogram of the gradient value frequency distribution of a network layer at a certain point during the training. Figure (a) counts all the parameters of the layer, while Figure (b) counts the randomly selected 1024 parameters. These gradient values belong to the 30th CONV layer of ResNet50 and were sampled during the training of ResNet50 on CIFAR-10.

To solve the above issue, we conduct an experiment to explore the relationship between a layer's parameter subsets and all its parameters. We observe the gradient value frequency distribution since the gradients are the most direct factor that determines the weight updates in the training process. As shown in Figure 3, we can observe that, for a network layer, the gradient distribution of the randomly selected weights subset is highly similar to the gradient distribution of all the weights. Therefore, we make the assumption that the updating pattern of parameter subsets can represent that of the whole layer. This characteristic provides us with the feasibility to design the *layer tailoring* technique. In specific, we randomly sample the weights from each layer to tailor all the layers to a uniform size (e.g., 1024) and fit the sampled weights into the predictor model. With our layer tailoring, the predictor can be shared by the layers with different sizes.

### 3.3 Workflow of the Lightweight Attention-based Predictor

The overview of the detailed predictor workflow is depicted in Figure 2(b). For each historical weight $W_l^j$ (sampled parameter of layer $l$ at timestamp $j \in \{0, 1, \cdots, t\}$), the predictor first encodes them into a query vector $Q_l^j$, a key vector $K_l^j$, and a value vector $V_l^j$, following Equation 1, in which $MLP_{k/q/v}(\cdot)$ denotes trainable three-layer Multi-Layer-Perceptrons.

$$K_l^j = MLP_k(W_l^j), Q_l^j = MLP_q(W_l^j), V_l^j = MLP_v(W_l^j), j \in \{0, 1, \cdots, t\} \tag{1}$$

Next, it computes the correlation score $a_l^j$ between the current time (i.e., iteration $t$) and each previous timestamp $j$. Specifically, it conducts dot-product between query vector at time $t$ (i.e., $Q_l^t$) with key vectors from timestamp $j$ (i.e., $K_l^j$), then gets a normalized attention score $\alpha_l^j$ by softmax across all previous timestamps, following Equation 2.

$$a_l^j = Q_l^t \cdot K_l^j, j \in \{0, 1, \cdots, t\} \tag{2}$$

$$\alpha_l^j = \frac{\exp\left(a_l^j\right)}{\sum_{i=0}^t \exp(a_l^i)}, j \in \{0, 1, \cdots, t\}$$

The value vector from each timestamp $j$ (i.e., $v_l^j$) is weighted by the attention score $\alpha_l^j$, and accumulated to the final context vector $C_l^t$, following Equation 3. Hence, the aggregated context vector keeps more information from those with higher attention scores while paying less attention to those with low scores.

$$C_l^t = \sum_{j=0}^t \alpha_l^j V_l^j \tag{3}$$

In the end, the prediction module $MLP_z(\cdot)$ (i.e., a three-layer MLP followed by a softmax) computes confident scores of freezing or not as a two-element vector. We choose a decision with higher confidence, as shown in Equation 4. If $d_l^t = 1$, we will freeze layer $l$ at timestamp $t$; otherwise, we continue its training at the moment. The predictor is lightweight in terms of both computing and memory overhead. Details of the model size and overheads are discussed in Appendix C.

$$d_l^t = Argmax(MLP_z(C_l^t)) \tag{4}$$

### 3.4 Training the Predictor

In order to not seize the computational resources during the model training, we decide to train the predictor offline and then use it during the training process instead of jointly training the predictor. To train the attention-based predictor, we need to generate a dataset ourselves. As mentioned in Section 3.2, we use the layer weights as input data and then apply a layer representational similarity-based method to label the input data.

**Layer Representational Similarity.** As training proceeds, the model layers are continuously being updated, and the representational similarity between a layer of the model being trained and the corresponding layer of the well-trained model will increase. The representational similarity indicates the feature extraction capability of a specific layer. We adopt a widely-used index Centered Kernel Alignment (CKA) to indicate the representational similarity of two layers (Kornblith et al., 2019). CKA is obtained by comparing the output feature maps of two layers under the same input image batch. It is calculated as:

$$CKA\left(X,Y\right) = \left\|Y^{\mathrm{T}}X\right\|_{\mathrm{F}}^{2} / \left(\left\|X^{\mathrm{T}}X\right\|_{\mathrm{F}} \left\|Y^{\mathrm{T}}Y\right\|_{\mathrm{F}}\right) \tag{5}$$

where $X$ and $Y$ are the output feature maps from two layers, and $\left\|\cdot\right\|_{\mathrm{F}}^{2}$ represents the square of the Frobenius norm of a matrix. A higher CKA value indicates that the two layers will output more similar feature maps with the same inputs. Further, if the CKA value of a layer stabilizes during the training process, we consider this layer is converged and ready to be frozen.

Figure 4 plots the CKA value curve of several CONV layers as training proceeds. As one can observe from the figure, after a period of fine-tuning, the layer's CKA stabilizes, which means that the feature extraction capability of those layers hardly changes even if the training continues. At this moment, freezing those layers will have little impact on accuracy. Moreover, the similarity with the well-trained model is not monotonically increasing but fluctuating, which correlates with our discussion in section 3.2 that it is not always the most recent model that has the most significant impact on the current model parameters. We can also observe from the figure that different layers require different training epochs to stabilize, and basically the layers in the front stabilize faster than the layers in the

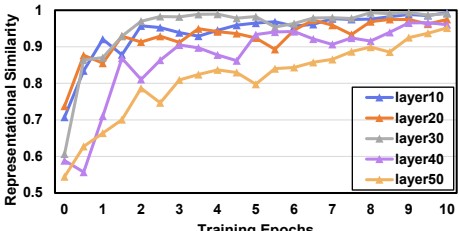

Figure 4: CKA variation curve as training proceeds. Result is obtained while training ResNet50 on CIFAR-10 and the ResNet50 is pre-trained on ImageNet.

back (e.g., layers 10 and 20 stabilize faster than layers 40 and 50). But it is interesting to see that the layer in the back is possible to stabilize faster than the front layer (e.g., layer 30 stabilizes faster than layers 10 and 20). This is due to the presence of residual connections in the ResNet architecture, which allows some of the later layers to behave like the earlier layers (Veit et al., 2016). These observations demonstrate the need for an adaptive layer freezing method, which is exactly what the SmartFRZ does, rather than forcing sequential freezing from front to back. Note that we cannot directly use this representational similarity-guided freezing because we cannot obtain the well-trained reference model in advance in practice.

**Generating the Training Dataset.** We first train a model on a certain dataset and use it as a well-trained reference model for CKA calculation. Then we train the same network on the same dataset again, in which we periodically calculate the CKA value between the well-trained model and the model under training for all the layers. With the CKA value, we are able to determine whether to freeze a layer or not. In this way, each piece of training data includes a sequence of historical weights of a tailored layer and a label indicating whether to freeze that layer or not at the moment of the latest record of this sequence.

We conduct an experiment to show the effectiveness of labeling the data through layer representational similarity. Specifically, we first train and obtain well-trained ResNet50 (He et al., 2016) and VGG11 (Simonyan & Zisserman, 2015) models on the CIFAR-100 (Krizhevsky & Hinton, 2009) dataset. Then we train another ResNet50 and VGG11 model with CIFAR-100 dataset again and freeze the network layers under the guidance of layer representational similarity during the training process. In this experiment, the freezing is conducted at the end of each epoch. Table 1 shows the experimental results of the training with and without layer freezing. As shown in the table, training

Table 1: Comparison of model accuracy and total computation of the training process using different methods: full training and layer representational similarity-guided freezing. Both models are pre-trained on the ImageNet dataset and then trained for 10 epochs on CIFAR-100.

| Method | ResNet50 | | VGG11 | |
|---|---|---|---|---|
| | Accuracy | Comp. (TFLOPs) | Accuracy | Comp. (TFLOPs) |
| Full Training | 81.68% | 12,360 | 74.95% | 22,950 |
| Similarity-Guided Freezing | 81.75% | 6,454 | 74.89% | 12,811 |

the networks using the representational similarity-guided freezing method can significantly reduce the computation cost (e.g., it saves 48% computation for training ResNet50) while maintaining the model accuracy, demonstrating the effectiveness of our similarity-guided labeling method and the correctness of the generated dataset.

## 4 EVALUATION

### 4.1 EXPERIMENTAL SETUP

In this section, we evaluate the proposed SmartFRZ framework in both computer vision and language domains. In CV domain, we use three representative CNN models ResNet50, VGG11, and MobileNetV2 (Sandler et al., 2018), and a vision transformer model DeiT-T (Touvron et al., 2021). And we use three widely-used datasets ImageNet (Deng et al., 2009), CIFAR-10, and CIFAR-100. In NLP domain, we fine-tune the pre-trained BERT-base model (Kenton & Toutanova, 2019) using two datasets MRPC (Dolan & Brockett, 2005) and CoLA (Warstadt et al., 2019) in GLUE benchmark (Wang et al., 2019). We conduct our experiments on two servers: i) a server with $8\times$ NVIDIA RTX 2080Ti GPUs is used for the experiments on ImageNet dataset and ii) an NVIDIA Tesla P100 GPU server is used in all other experiments. Both the predictor and the target networks are trained using the SGD optimizer with momentum. The training data is divided into batches with a size of 32. In CNN models, we freeze the BN layer together with its corresponding CONV layer to avoid unnecessary computation costs in back-propagation. The attention-based lightweight predictor is trained once on ImageNet using ResNet50 and then used in different models and datasets. The attention window size for the predictor is 30. And we tailor all the layers into the size of 1024 by random sampling to fit into the generic predictor. All the accuracy results in this paper are the average of 5 runs using different random seeds. And the overhead introduced by predictor is included in the results. A benchmark is defined as a model-dataset combination (e.g., VGG11+CIFAR-100).

### 4.2 MAIN RESULTS

**Fine-tuning.** Table 2 shows the results of different freezing methods (Linear Freezing (Brock et al., 2017) and AutoFreeze (Liu et al., 2021)). Linear freezing is a manually determined sequential freezing method. In our experiments, we set the number of frozen layers to increase according to the number of training epochs and let its total training time similar to that of SmartFRZ for fair comparison as it is highly rely on manually predefined freezing configurations. AutoFreeze is also a sequential freezing scheme and uses the variation of gradients norm as the metric to guide the freezing. More details of the hyperparameters of these two methods are provided in the Appendix B. SmartFRZ can significantly reduce computation cost and training time without compromising accuracy compared to full training. And it consistently outperforms Linear Freezing and AutoFreeze.

Compared to Linear Freezing, SmartFRZ provides up to 1.4% higher accuracy (VGG11 CIFAR-100), with 0.7% higher on average, while consuming similar time and energy. This is because Linear Freezing freezes layers in a fixed order at a pre-determined time, so some layers that are not well-trained might also be frozen. Moreover, linear freezing performs poorly on some benchmarks (i.e., VGG11 CIFAR-10, VGG11 CIFAR-100, and MobileNetV2 CIFAR-100). This is because linear freezing is highly dependent on a predefined freezing configuration, which does not adapt to different scenarios. For example, the training epochs required to converge is definitely different when training different model on different datasets with different complexity (e.g., ImageNet vs. MNIST), so the freeze configuration should also be different.

Compared to AutoFreeze, SmartFRZ reduces up to 20.4% time (MobileNetV2 CIFAR-10 benchmark) and 24.7% computation cost (VGG11 CIFAR-10 benchmark) while providing similar accu-

Table 2: Comparison of different freezing methods. All the models are pre-trained on ImageNet dataset. Then the CNN models and ViT model are trained for 10 epochs and 100 epochs to converge, respectively, using a cosine annealing learning rate scheduler according to the training epochs.

| Model | Method | CIFAR-10 | | | CIFAR-100 | | |
|---|---|---|---|---|---|---|---|
| | | Accuracy | Time (Second) | Computation (TFLOPs) | Accuracy | Time (Second) | Computation (TFLOPs) |
| ResNet50 | Full Training | 96.10%±0.12% | 2,594 | 12,360 | 81.68%±0.15% | 2,500 | 12,360 |
| | Linear Freezing | 96.05%±0.25% | 1,980 | 8,247 | 81.48%±0.29% | 1,844 | 7,085 |
| | AutoFreeze | 96.24%±0.38% | 2,424 | 8,716 | 81.30%±0.48% | 1,963 | 7,716 |
| | SmartFRZ (Ours) | 96.12%±0.19% | 1,955 | 8,122 | 81.73%±0.22% | 1,787 | 6,398 |
| VGG11 | Full Training | 93.36%±0.17% | 2,698 | 22,950 | 74.95%±0.11% | 2,703 | 22,950 |
| | Linear Freezing | 92.35%±0.19% | 1,587 | 10,291 | 73.26%±0.23% | 1,846 | 12,307 |
| | AutoFreeze | 93.08%±0.53% | 1,895 | 13,350 | 74.74%±0.36% | 2,027 | 13,946 |
| | SmartFRZ (Ours) | 93.58%±0.25% | 1,554 | 10,059 | 74.66%±0.29% | 1,831 | 12,121 |
| MobileNetV2 | Full Training | 94.15%±0.10% | 1,369 | 960 | 76.52%±0.21% | 1,371 | 960 |
| | Linear Freezing | 94.19%±0.26% | 977 | 567 | 75.59%±0.27% | 1,000 | 540 |
| | AutoFreeze | 94.26%±0.23% | 1,245 | 678 | 76.43%±0.42% | 1,248 | 652 |
| | SmartFRZ (Ours) | 94.03%±0.17% | 972 | 561 | 76.73%±0.20% | 986 | 532 |
| DeiT-T | Full Training | 97.48%±0.20% | 14,603 | 32,400 | 85.03%±0.28% | 14,628 | 32,400 |
| | Linear Freezing | 97.06%±0.23% | 8,760 | 16,290 | 83.89%±0.19% | 9,956 | 17,542 |
| | AutoFreeze | 97.35%±0.46% | 10,368 | 17,786 | 84.59%±0.37% | 11,154 | 18,710 |
| | SmartFRZ (Ours) | 97.65%±0.36% | 8,662 | 15,529 | 84.82%±0.25% | 9,599 | 16,636 |

racy. The reasons behind this are three-fold. First, SmartFRZ applies an attention-based predictor that can precisely determine the moment to freeze a layer. Second, AutoFreeze only freezes the layer where all the layers in front of it have been frozen. However, as discussed in Section 3.4, one layer might stabilize earlier than some of the layers in front of it. Therefore, forcing sequential freezing may result in wasting computation. Third, AutoFreeze incurs considerable overhead as it needs to compute the L2 Norm for all the gradients. Specifically, this overhead consumes 20%, 12%, 11% and 3% of the total training time while training ResNet50, MobileNetV2, DeiT-T, and VGG11, respectively. As a result, AutoFreeze only shows a marginal improvement in time consumption on the MobileNetV2 benchmark. We also evaluate the SmartFRZ framework in NLP domains and the experimental results are provided in the Appendix A.1.

**Training From Scratch.** In addition to fine-tuning, we also investigate the performance of our SmartFRZ framework in training a model from scratch. As shown in table 3, SmartFRZ saves training time by 22.4% on average without compromising accuracy compared to full training. SmartFRZ also consistently outperforms linear freezing and AutoFreeze in training from scratch. Compared to linear freezing, SmartFRZ provides 2% higher accuracy with 6.7% less time consumption on average. The performance of linear freezing degrades severely compared to fine-tuning. This is because the robustness of the raw model in the training from scratch experiment is much lower than that of the pre-trained model in the fine-tuning experiment. Linear freezing, which relies heavily on manually predefined freezing configuration, cannot handle this more complex training scenario. Compared to AutoFreeze, SmartFRZ provides a significantly 3.78% higher accuracy with 11.8% less time consumption. Besides the CNN model, we also evaluate the effectiveness of our SmartFRZ framework in training the vision transformer model DeiT-T from scratch, and the experimental results are shown in the Appendix A.2.

Table 3: Comparison of different freezing methods in training a model from scratch (160 epochs for ResNet50-CIFAR-10 and 100 epochs for ResNet50-ImageNet).

| Method | CIFAR-100 | | ImageNet | |
|---|---|---|---|---|
| | Accuracy | Time (Min.) | Accuracy | Time (Min.) |
| Full Training | 77.48%±0.14% | 711 | 76.89%±0.12% | 780 |
| Linear Freezing | 74.99%±0.19% | 615 | 75.45%±0.36% | 626 |
| AutoFreeze | 72.82%±0.25% | 630 | 74.06%±0.31% | 682 |
| SmartFRZ (Ours) | 77.64%±0.22% | 538 | 76.80%±0.26% | 621 |

## 4.3 MEMORY COST REDUCTION AND FREEZING PATTERN

Our SmartFRZ framework not only reduces training computation costs by freezing layers, but also reduces memory costs due to the reduction of intermediate data generated during back-propagation. Figure 5(a) shows the memory cost as training proceeds. We can observe from the figure that

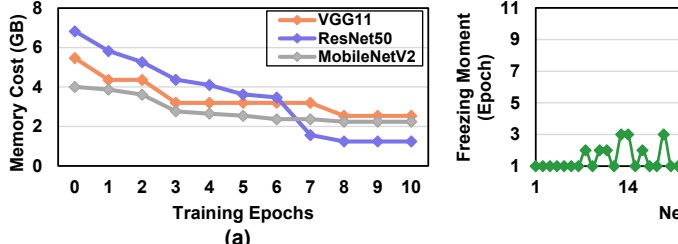

Figure 5: (a) Memory cost as training proceeds. Results are obtained by training VGG11, ResNet50, and MobileNetV2 on CIFAR-100 dataset. (b) The frozen moment for each layer while training ResNet50 on CIFAR-100.

Table 4: Comparison of the model accuracy when the attention window size for freezing prediction varies. The results are obtained using ResNet50 and VGG11 on CIFAR-100. The tailored layer size is set to 1024.

Table 5: Comparison of the model accuracy when the uniform tailored layer size varies. The results are obtained using ResNet50 and VGG11 on CIFAR-100. The attention window size is set to 30.

| Model | Window Size | | | |
|---|---|---|---|---|
| | 20 | 30 | 40 | 50 |
| ResNet50 | 81.17% | 81.73% | 81.75% | 82.11% |
| VGG11 | 74.45% | 74.66% | 75.11% | 74.67% |

| Model | Tailored Layer Size | | | |
|---|---|---|---|---|
| | 512 | 1024 | 2048 | 4096 |
| ResNet50 | 81.28% | 81.73% | 81.33% | 81.69% |
| VGG11 | 73.98% | 74.66% | 74.48% | 75.12% |

SmartFRZ can significantly save more than 80% memory for ResNet50 and about 50% memory for VGG11 and MobileNetV2. Considering the limited GPU memory capacity, our approach enlarges the potential of accommodating multiple DNN tasks within a single GPU. Figure 5(b) shows the freezing moment for each layer while training ResNet50 on CIFAR-100. As shown in the figure, although there are still some layers in the back getting frozen earlier than some front layers, Smart-FRZ tends to freeze the front layers earlier than the layers in the back. This is because the front layers mainly extract low-level features (e.g., detecting edges), while the back layers extract high-level features (Yosinski et al., 2014; Zeiler & Fergus, 2014). And the low-level feature extraction capability can be inherited from the pre-training process more easily.

### 4.4 SENSITIVITY STUDY AND OVERHEAD ANALYSIS

We investigate the effect of adopting different attention window sizes (i.e., the number of historical data used) to predict the freezing probability. Table 4 shows the accuracy when the attention window size varies. As one can observe, SmartFRZ consistently delivers a high model accuracy under different window sizes. It demonstrates the robustness of SmartFRZ in handling scenarios of different attention window sizes. This is because the attention mechanism adaptively selects valuable information within the long window ranges.

We also study how the uniform tailored layer size affects our SmartFRZ framework. As shown in table 5, our proposed SmartFRZ framework remains robust except for a very slight accuracy drop in accuracy (i.e., less than 1%) when the layer size is 512. And the reason behind this is that if the tailored layer size is too small, the sampled parameters might be hard to represent the status of the whole layer's parameters. Therefore, we tailor the layer to the size of at least 1024, which samples richer layer information while yielding negligible overheads (Details discussed in Section B).

## 5 CONCLUSION

This work considers an efficient training technique layer freezing on both fine-tuning and training from scratch scenarios. We propose an attention-based layer freezing framework SmartFRZ. The key idea is to leverage the training histories to predict the freezing probability of a layer during the training process. Our attention-based layer freezing predictor can learn comprehensive information about the network layer. The predictor is trained offline on a dataset with layer weights as input data which is labeled by a layer representation similarity-based method. Our extensive experiments demonstrate that SmartFRZ significantly saves training time and computation by automatically freezing layers without compromising accuracy.

ACKNOWLEDGMENTS

The authors would like to thank the anonymous reviewers for their constructive feedback and suggestions. This work is supported in part by NSF grants #2011146, #2154973, #1725657, and #1909172.

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

APPENDIX

## A  EXTENDED EXPERIMENTAL RESULTS

### A.1  NLP TASKS USING BERT-BASE

In this section, we provide the experimental results of different freezing approaches in NLP tasks. As shown in Table 6, SmartFrz is able to significantly reduce fine-tuning time without accuracy loss. Moreover, SmartFrz delivers an average of 0.86% higher accuracy than linear freezing with similar training time. Compared to AutoFreeze, SmartFrz provides 0.68% higher accuracy while consuming 10.5% less time on average. These results demonstrate the superiority of SmartFrz over linear freezing and AutoFreeze.

Table 6: Comparison of different freezing methods in NLP domain. The results are obtained by fine-tuning 5 epochs of pretrained BERT-base on two datasets MRPC and CoLA, respectively.

| Method | MRPC | | CoLA | |
|---|---|---|---|---|
| | Accuracy | Time (Second) | Accuracy | Time (Second) |
| Full Training | 86.51%±0.24% | 316 | 56.65%±0.18% | 723 |
| Linear Freezing | 85.80%±0.21% | 221 | 55.90%±0.29% | 429 |
| AutoFreeze | 86.02%±0.36% | 266 | 56.05%±0.21% | 431 |
| SmartFRZ (Ours) | 86.76%±0.25% | 215 | 56.66%±0.22% | 423 |

### A.2  TRAINING FROM SCRATCH EXPERIMENTS USING DEIT-T MODEL

Table 7: Comparison of different freezing methods in training DeiT-T from scratch on ImageNet dataset (300 epochs).

| Method | Accuracy | Time (Minute) |
|---|---|---|
| Full Training | 72.21%±0.12% | 2,190 |
| Linear Freezing | 71.19%±0.24% | 1,817 |
| AutoFreeze | 71.55%±0.48% | 1,815 |
| SmartFRZ (Ours) | 72.06%±0.29% | 1,801 |

In this section, we provide the experimental results of training the vision transformer model DeiT-T from scratch on ImageNet dataset. As one can observe from Table 7, SmartFRZ delivers higher accuracy (0.69% on average) than linear freezing and AutoFreeze with the lowest training time.

## B  CONFIGURATIONS OF LINEAR FREEZING AND AUTOFREEZE BASELINES

To show the fairness of the experiments, in this section, we provide the details of the configurations of linear freezing (Brock et al., 2017) and AutoFreeze (Liu et al., 2021).

**Linear Freezing.** Linear freezing employs a layer-wise cosine annealing learning rate schedule without restarts, where the first layer's learning rate is reduced to zero partway through training (at $t_0$), and each subsequent layer's learning rate is annealed to zero thereafter. Once a layer's learning rate reaches zero, it is frozen and will not be updated afterward. The learning rate $\alpha_i$ of layer $l_i$ (whose learning rate anneals to zero at $t_i$) at iteration $t$ is calculated as:

$$\alpha_i(t) = 0.5 * \alpha_i(0)(1 + \cos \pi t / t_i) \qquad (6)$$

There is one hyperparameter in linear freezing, $t_0$, which defines which epoch the learning of the first layer reaches zero. The value of $t_i$ is linearly spaced between $t_0$ and the total number of epochs. We explore the impact of the value of $t_0$. Table 8 shows the results with different configurations of $t_0$. Following the setting in the original paper, we vary the number of total training epochs for each configuration to make the total training time of each configuration almost the same. As shown in the table, with similar training time, setting $t_0$ to 0.5 provides the highest accuracy, which is also

Table 8: Hyperparameter sweep at different $t_0$ values in linear freezing (Brock et al., 2017). The results are obtained by fine-tuning ResNet50 on CIFAR-10 dataset.

|  | Accuracy | Time (Second) |
|---|---|---|
| $t_0 = 0.3$ | 95.62% | 1,989 |
| $t_0 = 0.4$ | 95.81% | 1,986 |
| $t_0 = 0.5$ | 96.05% | 1,980 |
| $t_0 = 0.6$ | 95.97% | 2,015 |

recommended by the original paper. To ensure fair comparisons, we have selected the best results of linear freezing as a baseline to compare against our method.

**AutoFreeze.** AutoFreeze periodically freezes layers whose gradients norm change rate is low. There are two hyperparameters in the AutoFreeze framework. The first one is the number of evaluation intervals $M$, which indicates the frequency of freezing layers. For example, 4 intervals/epoch means that AutoFreeze evaluates the gradient norm change rate for each active layer and freezes the layers with a low gradient norm change rate 4 times per epoch. The second hyperparameter is the threshold $N$ to determine whether a layer is converged or not. If a layer's gradient norm change rate is in the bottom $N^{th}$ percentile of all the active layers, it is considered converged and ready to be frozen (there is another requirement that a layer can be frozen until all the layers in front of it have been frozen).

Table 9: Hyperparameter sweep at different number of evaluation intervals per epoch in AutoFreeze (Liu et al., 2021). The results are obtained by fine-tuning ResNet50 on CIFAR-10 dataset.

|  | Accuracy | Time (Second) |
|---|---|---|
| $M = 2$ | 96.19% | 2,536 |
| $M = 3$ | 96.28% | 2,487 |
| $M = 4$ | 96.24% | 2,424 |
| $M = 5$ | 95.69% | 2,378 |

In our experiments, we follow the recommendation from original paper (Liu et al., 2021) and set the $N$ to 50%. For a fair comparison, we further study how the number of evaluation intervals per epoch (i.e., $M$) affects the accuracy and training time. We conduct the experiments by varying the intervals per epoch from 2 to 5 since (Liu et al., 2021) finds the trade-off between accuracy and training time is balanced for a range of values (2 to 5 intervals/epoch). Table 9 shows the experimental results, and it can be observed from the table that sets the $M$ to 4 can maintain the accuracy with high training time reduction. To this end, in our main paper, we set the $N$ to 50% and $M$ to 4 to get the best results of AutoFreeze.

## C  OVERHEAD ANALYSIS

In terms of predictor size, our lightweight predictor consists of only one $MLP_k(\cdot)$, one $MLP_q(\cdot)$, one $MLP_v(\cdot)$, and one $MLP_z(\cdot)$, as shown in Figure 2. In specific, the $MLP_k(\cdot)$, $MLP_q(\cdot)$, and $MLP_v(\cdot)$ in equation 1, which are used for encoding weight histories, are 3-layer Multi-Layer-Perceptrons (input size as 1024, hidden sizes as 256, and output sizes as 64). We adopt one layer of attention (i.e., without stacking multiple attention layers as BERT Kenton & Toutanova (2019)) with one single head (i.e., encoding one single group of key, query, and value vectors with one group of $MLP_k(\cdot)$, $MLP_q(\cdot)$, and $MLP_v(\cdot)$ for each timestamp) in our design. The $MLP_z(\cdot)$ in equation 4, which is used for computing confident scores of freezing or not, is a 3-layer Multi-Layer-Perceptron (input size as 64, hidden sizes as 32, and output sizes as 2). To this end, the whole predictor consists of four 3-layer MLPs. The predictor size is about 4 MB, which is negligible compared to the training overheads. Moreover, since we tailor the layer by subsampling layer weights to a fixed-size vector (e.g., a size of 1024), SmartFRZ only requires hundreds of KB to store the whole model once (e.g., about 200 KB for ResNet50). As such, SmartFRZ only requires about 6 MB of memory to store the historical weights for freezing prediction even with a large attention window of size 30.

The computation cost for one inference is 0.12 GFLOPs if we set the attention window size to 30 and the tailored layer size to 1024. So, it only consumes negligible less than 0.1% of total training time. Noticeably, the computation complexity of our predictor is linear-scaled to the sequence length instead of quadratic-scaled since it handles the sequence classification task (which requires all-to-one attention) instead of the sequence-to-sequence translation task (which requires all-to-all attention). The encoding stage (i.e., equation 1) encodes each time stamp, yielding $\mathcal{O}(t)$ complexity. The aggregating stage (i.e., equation 2 equation 3) calculates weights from the current timestamp to sequences of timestamps as a one-to-all procedure, and it causes $\mathcal{O}(t)$ as well. Furthermore, the last MLP classification steps only cause $\mathcal{O}(1)$. In summary, our predictor has linear complexity ($\sim \mathcal{O}(t)$) in terms of sequence length instead of quadratic complexity ($\sim \mathcal{O}^2(t)$). Moreover, once a layer is frozen, we do not need to predict the freezing decision for that layer anymore and can discard its historical weights. So the computation cost and memory cost introduced by the predictor will decrease as training proceeds.

