# OpenReview forum: "SmartFRZ: An Efficient Training Framework using Attention-Based Layer Freezing"
_ICLR.cc/2023/Conference — ICLR 2023 notable top 25%_

### Official Review · Reviewer_QAQ7 · 2022-10-25

**Confidence:** 4
**Correctness:** 4
**Technical Novelty And Significance:** 3
**Empirical Novelty And Significance:** 3
**Recommendation:** 8

**Clarity, Quality, Novelty And Reproducibility:**

Yes, the presentation is good. The analyses of the experiments and ablations are adequate and solid.

**Strength And Weaknesses:**

Strength:
The motivation is clear and the topic is useful. The proposed predictor can dynamically choose which layer to be frozen to enable training efficiency and memory saving.

Weakness:
The predictor also consumes the memory during normal training. Seems that there are no numerical results about this aspect. Do all reductions of memories or time reported in the paper have taken the predictor part into accounting？
How do you choose the structure and config of the predictor according to different models that have very various parameters?
Training from scratch experiments did not include the results on the ImageNet dataset. Are there any reasons for this?


**Summary Of The Paper:**

The paper proposed an interesting topic of adaptively freezing the layers during training to save training time and memory.  The proposed predictor output attention for each layer by integrating the history layer information during training. The training and freezing are alternatively performed on layers of models.

**Summary Of The Review:**

Although there are some weaknesses in this work, I still think that this work is interesting and can be an easy-to-use tool in practice.  The authors are welcome to release the code and describe the additional memory cost brought by this predictor. And if possible, results on ImageNet are also convincing.

---

> ### Author Response · Authors · 2022-11-18
> **Author Response to Reviewer QAQ7 (Part 2/2)**
>
> ### **Q4. Results on ImageNet.**
>
> **Thank you for the valuable suggestion.**
>
> To further validate the effectiveness of our SmartFRZ, we add more results including the training from scratch results on **ImageNet** on both **ResNet50** and the **vision transformer model (DeiT)**. Besides, we also add the results of the **BERT model for NLP** downstream tasks.
>
> The results show that our SmartFRZ framework consistently outperforms linear freezing and AutoFreeze in training from scratch experiments on the ImageNet dataset as well as the BERT model for NLP downstream tasks.
>
> * **Training ResNet50 from scratch on ImageNet**
>
> We add the experiments of training ResNet50 from scratch on the ImageNet dataset. The results are provided in Table 3 in Section 4.2 (Page 8) in the revised paper.
>
> We also put the experimental results in Table R.22 here.
>
> >**Table R.22: Comparison of different freezing methods in training ResNet50 from scratch on ImageNet dataset (100 epochs).**
> | Method      	| 	Accuracy 	|   Time (Minute)        	|
> |-----------------|:----------------:|---------------|
> | Full Training   | 76.89\%$\pm$0.12\% | 780       	|
> | Linear Freezing | 75.45\%$\pm$0.36\% | 626       	|
> | AutoFreeze  	| 74.06\%$\pm$0.31\% | 682       	|
> | **SmartFRZ (Ours)** | **76.80\%$\pm$0.26\%** | **621**     	|
>
> And we can observe from the table that SmartFRZ delivers significantly higher accuracy, with 1.35% higher than linear freezing and 2.74% higher than AutoFreeze. SmartFRZ also consumes 8.9% less time than AutoFreeze.
>
> * **Training vision transformer model (DeiT-T) from scratch on ImageNet**
>
> We evaluate our method in training DeiT-T from scratch on the ImageNet dataset. The results of fine-tuning experiments are provided in Appendix A.2 and Table 7 (Page 13) in the revised paper. We also put here the results mentioned above, as shown in Table R.23.
>
> >**Table R.23: Comparison of different freezing methods in training DeiT-T from scratch on ImageNet dataset (300 epochs).**
> | Method                          	| Accuracy       	| Time (Minute) |
> |-------------------------------------|--------------------|---------------|
> | Full Training                       | 72.21\%$\pm$0.12\% | 2,190     	|
> | Linear Freezing                 	| 71.19\%$\pm$0.24\% | 1,817     	|
> | AutoFreeze                      	| 71.55\%$\pm$0.48\% | 1,815     	|
> | **SmartFRZ (Ours)** | **72.06\%$\pm$0.29\%** | **1,801**     	|
>
> Table R.23 shows the results of training the vision transformer model DeiT-T from scratch on the ImageNet dataset. As one can observe, SmartFRZ delivers higher accuracy (0.69% on average) than linear freezing and AutoFreeze with the lowest training time.
>
> * **BERT for NLP Tasks**
>
> We also evaluate our method on the NLP domain using the BERT-base model. The results are provided in Appendix A.1 and Table 6 (Page 13) in the revised paper.
>
> We also put the results in Table R.24 here.
>
> >**Table R.24: Comparison of different freezing methods in the NLP domain. The results are obtained by fine-tuning (5 epochs) BERT-base on two datasets, MRPC and CoLA.**
> | Method                              |    	MRPC    	|          	 |        CoLA    	|           	|
> |-------------------------------------|:------------------:|---------------|:------------------:|---------------|
> |                                 	| Accuracy       	| Time (Second) | Accuracy       	| Time (Second) |
> | Full Training                   	| 86.51\%$\pm$0.24\% | 316       	| 56.65\%$\pm$0.18\% | 723       	|
> | Linear Freezing                 	| 85.80\%$\pm$0.21\% | 221       	| 55.90\%$\pm$0.29\% | 429       	|
> | AutoFreeze                      	| 86.02\%$\pm$0.36\% | 266       	| 56.05\%$\pm$0.21\% | 431       	|
> | **SmartFRZ (Ours)** | **86.76\%$\pm$0.25\%** | **215**       	| **56.66\%$\pm$0.22\%** | **423**       	|
>
> As shown in Table R.24, SmartFrz is able to significantly reduce fine-tuning time without accuracy loss. Moreover, SmartFrz delivers an average of 0.86% higher accuracy than linear freezing with similar training times. Compared to AutoFreeze, SmartFrz provides 0.68% higher accuracy while consuming 10.5% less time on average. These results demonstrate the superiority of SmartFrz over linear freezing and AutoFreeze.

---

> ### Author Response · Authors · 2022-11-18
> **Author Response to Reviewer QAQ7 (Part 1/2)**
>
> **We would like to thank the reviewer for the positive feedback and valuable suggestions. We appreciate the reviewer's acknowledgment that our proposed work is interesting and can be an easy-to-use tool in practice. We carefully address all the reviewer’s questions and revise the paper accordingly. We hope our response can help clarify the reviewer's questions.**
>
> ---
>
> ### **Q1.  The overhead of the predictor**
>
> Thank you for the question.
>
> Yes, the reported time and memory costs include all the overhead introduced by the predictor. We also include a detailed overhead analysis in the revised paper (Appendix C, Page 14).
>
> We would like to kindly clarify that our attention-based predictor is a lightweight model. And both our proposed layer tailoring technique and the confined attention window make the overhead of the predictor negligible (consumes only **10 MB** memory in total and less than 0.1% of total training time).
>
> In specific, our lightweight predictor consists of only one $MLP_k(\cdot)$, one $MLP_q(\cdot)$, one $MLP_v(\cdot)$, and one $MLP_z(\cdot)$, as shown in Figure 2, Page 4 in the main paper. The $MLP_k(\cdot)$, $MLP_q(\cdot)$, and $MLP_v(\cdot)$ in equation 1, which are used for encoding weight histories, are 3-layer Multi-Layer-Perceptrons (input size as 1024, hidden sizes as 256, and output sizes as 64). We adopt one layer of attention (i.e., without stacking multiple attention layers as BERT) with one single head (i.e., encoding one single group of key, query, and value vectors with one group of $MLP_k(\cdot)$, $MLP_q(\cdot)$, and $MLP_v(\cdot)$ for each timestamp) in our design. The $MLP_z(\cdot)$ in equation 5, which is used for computing confident scores of freezing or not, is a 3-layer Multi-Layer-Perceptron (input size as 64, hidden sizes as 32, and output sizes as 2). To this end, the whole predictor consists of four 3-layer MLPs. The predictor size is about **4 MB**, which is negligible compared to the training overheads.
>
> Moreover, since we tailor the layer by subsampling layer weights to a fixed-size vector (e.g., a size of 1024), SmartFRZ only requires hundreds of KB to store the whole model once (e.g., about 200 KB for ResNet50). As such, SmartFRZ only requires about **6 MB** of memory to store the historical weights for freezing prediction, even with a large attention window of size 30.
>
> Moreover, once a layer is frozen, we do not need to predict the freezing decision for that layer anymore and can discard its historical weights. So the computation cost and memory cost introduced by the predictor will decrease as training proceed.
>
> ---
>
> ### **Q2. How to choose the structure and config of the predictor for different models that have very various parameters?**
>
> Thank you for your question.
>
> **We would like to kindly clarify that we use the same predictor to guide the layer freezing for different datasets and networks, which is one major advantage of our predictor.**
>
> The predictor is trained offline once to learn the generic convergence pattern along the training history, thanks to the superiority of the attention mechanism. The predictor training only needs to use a single generated dataset (we only use ImageNet + ResNet50 to generate it). The trained predictor can be **generalized across different models and datasets without fine-tuning**. When using the trained predictor for guiding the layer freezing during the actual network training process, it does not require extra computation costs to train the predictor. Please also refer to Section 3.4.
>
> Moreover, to make our predictor lightweight, we also propose a layer tailoring technique, allowing our predictor to be shared across all layers that have different sizes. Please also refer to Section 3.2 layer tailoring part.
>
> We also provide the sensitivity analysis for the predictor configurations in Table 4 and Table 5 in the main paper.
>
> ---
>
> ### **Q3. Code release.**
>
> We would like to thank you again for the acknowledgment that our proposed work can be an easy-to-use tool in practice. We upload the code as supplementary materials. And we will also officially release the code upon the acceptance of the paper.

---

> ### Author Response · Authors · 2022-12-01
> **Author Response to Reviewer QAQ7**
>
> Dear Reviewer QAQ7,
>
> Thanks for your time and reviewing efforts to help improve our work! We appreciate your positive rating and thoughtful comments.
>
> We provide suggested results in the authors' response, such as the numeric results on overhead and experiments on more models and datasets. We hope our responses have answered your questions.
>
> Best,
>
> Authors

---

> > ### Comment · Reviewer_QAQ7 · 2022-12-04
> > **My concerns have been addressed.**
> >
> > Thanks for the detailed responses, I think it is clear to me now.

---

> > > ### Author Response · Authors · 2022-12-04
> > > **Thanks for the positive feedback**
> > >
> > > We would like to thank you again for your time and your positive comments. It is a great confirmation of our work.

---

### Official Review · Reviewer_Uzai · 2022-10-30

**Confidence:** 3
**Clarity, Quality, Novelty And Reproducibility:** It is well written.
**Correctness:** 3
**Technical Novelty And Significance:** 3
**Empirical Novelty And Significance:** 2
**Recommendation:** 6

**Strength And Weaknesses:**

Strength:

* A valid efficient training method towards cheaper DNN training.

Weakness or Questions:

* I doubt the necesarity of using attention-based model to predict the layers. Actually, attention-based model suffers from the quadractic complexity as compared to the local feature extractor, CNNs. So use Attention model to guide CNN training looks overkill to me. Instead, we should use small models to predict much larger models' freezing schedule.

* How is the overhead of such a attention based model? Since it will be repeatedly used for prediction, thus it must have to be negligible overhead otherwise I cannot see the point.

* The most direct baseline is to reduce the layer numbers and compare with your method, and randomly drop some layers during training process. I am not sure whether your predicted schedule beats the random one.

* Also, the exps are mostly conducted on CNNs, how about ViTs? We should natually reuse the attention based model as both predictors and main contributors to the model performance in the ViTs.

**Summary Of The Paper:**

This paper propose a dynamic layer freezing technique during DNN training. Specifically, they adopt attention-based predictor to predict which layer should be freezed. The predictor is jointly applyed to each timestamp.

**Summary Of The Review:**

In summary, I think that this paper provide a decent point of view for efficient training method. But the idea is not very novel as there are also many other works working on freezing layers. But the proposed methods beat several baselines.

Also, it is not clear whether it is worth to leverage attention model to help CNN training. It should be reversed IMO.

I suggests that the author could consider adding more ViT-based exps, where their attention can be naturally server as predictors.

If that works, I would increase my score as it shows that your methods can be generalized to ViT models.

---

> ### Author Response · Authors · 2022-11-18
> **Author Response to Reviewer Uzai (Part 3/3)**
>
> ### **Q3.The generalizability of our predictor to the vision transformer model. (cont.)**
>
> * **BERT for NLP Tasks**
>
> We evaluate our method on the NLP domain using the BERT-base model. The results are provided in Appendix A.1 and Table 6 (Page 13) in the revised paper.
>
> We also put the results in Table R.20 here.
>
> >**Table R.20: Comparison of different freezing methods in the NLP domain. The results are obtained by fine-tuning (5 epochs) BERT-base on two datasets, MRPC and CoLA.**
> | Method                              |    	MRPC    	|          	 |        CoLA    	|           	|
> |-------------------------------------|:------------------:|---------------|:------------------:|---------------|
> |                                 	| Accuracy       	| Time (Second) | Accuracy       	| Time (Second) |
> | Full Training                   	| 86.51\%$\pm$0.24\% | 316       	| 56.65\%$\pm$0.18\% | 723       	|
> | Linear Freezing                 	| 85.80\%$\pm$0.21\% | 221       	| 55.90\%$\pm$0.29\% | 429       	|
> | AutoFreeze                      	| 86.02\%$\pm$0.36\% | 266       	| 56.05\%$\pm$0.21\% | 431       	|
> | **SmartFRZ (Ours)** | **86.76\%$\pm$0.25\%** | **215**       	| **56.66\%$\pm$0.22\%** | **423**       	|
>
> As shown in Table R.20, SmartFrz is able to significantly reduce fine-tuning time without accuracy loss. Moreover, SmartFrz delivers an average of 0.86% higher accuracy than linear freezing with similar training times. Compared to AutoFreeze, SmartFrz provides 0.68% higher accuracy while consuming 10.5% less time on average. These results demonstrate the superiority of SmartFrz over linear freezing and AutoFreeze.
>
> * **Training ResNet50 from scratch on ImageNet**
>
> We also add the experiments of training ResNet50 from scratch on the ImageNet dataset. The results are provided in Table 3 in Section 4.2 (Page 8) in the revised paper.
>
> We also put the experimental results in Table R.21 here.
>
> >**Table R.21: Comparison of different freezing methods in training ResNet50 from scratch on ImageNet dataset (100 epochs).**
> | Method      	| 	Accuracy 	|   Time (Minute)        	|
> |-----------------|:----------------:|---------------|
> | Full Training   | 76.89\%$\pm$0.12\% | 780       	|
> | Linear Freezing | 75.45\%$\pm$0.36\% | 626       	|
> | AutoFreeze  	| 74.06\%$\pm$0.31\% | 682       	|
> | **SmartFRZ (Ours)** | **76.80\%$\pm$0.26\%** | **621**     	|
>
> And we can observe from the table that SmartFRZ delivers significantly higher accuracy, 1.35% higher than linear freezing and 2.74% higher than AutoFreeze. SmartFRZ also consumes 8.9% less time than AutoFreeze.
>
> ---
>
> ### **Q4. Reuse the attention-based model as both predictors and main contributors to the model performance in the ViTs.**
>
> **Thank you for this inspiring suggestion.**
>
> We would like to share some of our thoughts about this point.
>
> Reusing the attention model for both layer freezing prediction and the main task (e.g., vision feature extraction) is a promising direction that is definitely worth further exploration.
>
> Currently, there are still some challenges. The layer-freezing-oriented attention and the ViT-oriented attention take different types of inputs and have different goals at the task level.
>
> **The ViT-oriented attention focuses more on data.** It takes layer outputs (i.e., feature maps) as inputs of the attention, computes attention scores between patches from the inputs, and then produces the current layer's output feature maps.
>
> **The layer-freezing-oriented attention focuses more on weights.** It takes layer weights as inputs, and computes attention scores between layer histories from different timestamps, then produces the weight convergence pattern as outputs.
>
> Therefore, it is desirable to come up with a method to effectively combine those different objectives. We would like to thank you again for this interesting idea, and we will investigate it in future studies.

---

> ### Author Response · Authors · 2022-11-18
> **Author Response to Reviewer Uzai (Part 2/3)**
>
> ### **Q2.Comparison with randomly dropping layers during freezing.**
>
> We conduct experiments to compare our SmartFRZ framework with random freezing. For a fair comparison, we make the training time of random freezing similar to that of SmartFRZ. As shown in Table R.17, SmartFRZ delivers 1.16% higher accuracy than random freezing with similar training time.
>
> >**Table R.17: Comparison of SmartFRZ and random freezing on fine-tuning experiments (10 epochs for ResNet50 and 100 epochs for DeiT-T).**
> | Model	| Method      	| CIFAR-10 |           	| CIFAR-100 |           	|
> |----------|-----------------|----------|---------------|-----------|---------------|
> |      	|             	| Accuracy | Time (Second) | Accuracy  | Time (Second) |
> | ResNet50 | Random Freezing | 95.34%   | 1,988     	| 80.61%	| 1,812     	|
> |      	| SmartFRZ    	| 96.12%   | 1,955     	| 81.73%	| 1,787     	|
> | DeiT-T   | Random Freezing | 96.27%   | 8,683     	| 83.46%	| 9,643     	|
> |      	| SmartFRZ    	| 97.65%   | 8,662     	| 84.82%	| 9,599     	|
>
> ---
>
> ### **Q3.The generalizability of our predictor to the vision transformer model**
> Thank you for the valuable suggestion.
>
> As suggested by the reviewer, we add more results for the transformer-based model, such as the **vision transformer model (DeiT)** and the **BERT model for NLP**. Besides, we also add **training from scratch on ImageNet using ResNet50**.
>
> **The results demonstrate that our SmartFRZ also performs well on transformer-based models such as DeiT and BERT.**
>
> * **Vision transformer model (DeiT-T)**
>
> We evaluate our method on DeiT-T for fine-tuning and training from scratch scenarios. The results of fine-tuning experiments are added to Table 2 in Section 4.2 (Page 8) in our revised paper. The results of training from scratch experiments are provided in Appendix A.2 and Table 7 (Page 13) in the revised paper.
>
> We also put here the results mentioned above, as shown in Table R.18 and Table R.19.
>
> >**Table R.18: Comparison of different freezing methods in fine-tuning a vision transformer model DeiT-T. It is pre-trained on the ImageNet dataset and then fine-tuned for 100 epochs to converge.**
> | Method      	|      CIFAR-10  	|      	|         	|  	CIFAR-100 	|          |         	|
> |-----------------|:------------------:|----------|-------------|:------------------:|----------|-------------|
> |             	| Accuracy       	| Time 	| Computation | Accuracy       	| Time 	| Computation |
> |             	|                    | (Second) | (TFLOPs)	|                	| (Second) | (TFLOPs)	|
> | Full Training   | 97.48\%$\pm$0.20\% | 14,603   | 32,400  	| 85.03\%$\pm$0.28\% | 14,628   | 32,400  	|
> | Linear Freezing | 97.06\%$\pm$0.23\% | 8,760	| 16,290  	| 83.89\%$\pm$0.19\% | 9,956	| 17,542  	|
> | AutoFreeze  	| 97.35\%$\pm$0.46\% | 10,368   | 17,786      | 84.59\%$\pm$0.37\% | 11,154   | 18,710  	|
> | **SmartFRZ (Ours)** | **97.65\%$\pm$0.36\%** | **8,662**    | **15,529**      | **84.82\%$\pm$0.25\%** | **9,599**	| **16,636**  	|
>
> >**Table R.19: Comparison of different freezing methods in training DeiT-T from scratch on ImageNet dataset (300 epochs).**
> | Method                          	| Accuracy       	| Time (Minute) |
> |-------------------------------------|--------------------|---------------|
> | Full Training                       | 72.21\%$\pm$0.12\% | 2,190     	|
> | Linear Freezing                 	| 71.19\%$\pm$0.24\% | 1,817     	|
> | AutoFreeze                      	| 71.55\%$\pm$0.48\% | 1,815     	|
> | **SmartFRZ (Ours)** | **72.06\%$\pm$0.29\%** | **1,801**     	|
>
> We can observe from Tables R.18 and R.19 that our SmartFRZ framework consistently outperforms linear freezing and AutoFreeze.
>
> Table R.18 shows the results of fine-tuning the vision transformer model DeiT-T. Compared to linear freezing, it delivers 0.72% higher accuracy on average while consuming slightly less training time. Compared to AutoFreeze, it delivers 0.27% higher accuracy on average with 15.2% less training time.
>
> Table R.19 shows the results of training the vision transformer model DeiT-T from scratch on the ImageNet dataset. As one can observe, SmartFRZ delivers higher accuracy (0.69\% on average) than linear freezing and AutoFreeze with the lowest training time.

---

> ### Author Response · Authors · 2022-11-18
> **Author Response to Reviewer Uzai (Part 1/3)**
>
> **We appreciate the valuable comments and suggestions from the reviewer. We carefully address all the reviewer’s questions and revise the paper accordingly. We hope our response can help alleviate the reviewer's concern.**
>
> ---
>
> ### **Q1. The necessity to use a large attention-based predictor and its quadratic complexity issue. The overhead of predictor.**
>
> Thank you for this question.
>
> **We humbly think the reviewer might confuse about the size of our attention-based predictor. Based on that, the reviewer has a major concern about the necessity of using an attention-based predictor and its overheads.**
>
> We would like to kindly clarify that **our attention-based predictor** is a **lightweight model**. And both our proposed layer tailoring technique and the confined attention window make the overhead of the predictor **negligible** (consumes only **10 MB** memory and **less than 0.1%** of total training time).
>
> * **For the predictor size**
>
> Our lightweight predictor consists of only one $MLP_k(\cdot)$, one $MLP_q(\cdot)$, one $MLP_v(\cdot)$, and one $MLP_z(\cdot)$, as shown in Figure 2, Page 4 in the main paper. The $MLP_k(\cdot)$, $MLP_q(\cdot)$, and $MLP_v(\cdot)$ in equation 1, which are used for encoding weight histories, are 3-layer Multi-Layer-Perceptrons (input size as 1024, hidden sizes as 256, and output sizes as 64). We adopt one layer of attention (i.e., without stacking multiple attention layers as BERT) with one single head (i.e., encoding one single group of key, query, and value vectors with one group of $MLP_k(\cdot)$, $MLP_q(\cdot)$, and $MLP_v(\cdot)$ for each timestamp) in our design. The $MLP_z(\cdot)$ in equation 5, which is used for computing confident scores of freezing or not, is a 3-layer Multi-Layer-Perceptron (input size as 64, hidden sizes as 32, and output sizes as 2). To this end, the whole predictor consists of four 3-layer MLPs. The predictor size is about **4 MB**, which is negligible compared to the training overheads.
>
> Moreover, since we tailor the layer by subsampling layer weights to a fixed-size vector (e.g., a size of 1024), SmartFRZ only requires hundreds of KB to store the whole model once (e.g., about 200 KB for ResNet50). As such, SmartFRZ only requires about **6 MB** of memory to store the historical weights for freezing prediction, even with a large attention window of size 30.
>
> The computation cost for one inference is 0.12 GFLOPs if we set the attention window size to 30 and the tailored layer size to 1024 (as we used in our paper). So, it only has a negligible overhead on total training time (**less than 0.1%**).
>
> * **For complexity**
>
> In terms of complexity, while the attention mechanism usually causes quadratic complexity in the sequence-to-sequence tasks, our predictor handles sequence classification (i.e., sequence-to-one) tasks instead of the sequence-to-sequence task, which yields linear-scaled (to the sequence length) complexity. The encoding stage (i.e., Equation 1) encodes each time stamp, yielding $O(t)$ complexity. The aggregating stage (i.e., Equation 2&3) calculates weights from the current timestamp to sequences of timestamps as a one-to-all procedure, and it causes $O(t)$ as well. Furthermore, the last MLP classification steps only cause $O(1)$. In summary, our predictor has linear complexity ($\sim$$O(t)$) in terms of sequence length instead of quadratic complexity ($\sim$$O(t^2)$).
>
> Moreover, once a layer is frozen, we do not need to predict the freezing decision for that layer anymore and can discard its historical weights. So the computation cost and memory cost introduced by the predictor will decrease as training proceed.
>
> We make our overhead analysis clearer in the revised paper (Appendix C, Page 14).

---

> ### Author Response · Authors · 2022-12-01
> **Author Response to Reviewer Uzai**
>
> Dear Reviewer Uzai,
>
> Thanks for your time and reviewing efforts! We appreciate your constructive comments.
>
> We provide suggested results in the authors' response, such as the numeric results of our predictor's overhead, experiments on vision transformer models and more datasets to show the generalizability, and comparison with randomly freezing layers. We hope our responses have answered your questions. It would be our great pleasure if you would consider updating your review or score.
>
> Best,
>
> Authors

---

> ### Author Response · Authors · 2022-12-08
> **Looking forward to your feedback**
>
> Dear Reviewer Uzai,
>
> We would like to sincerely thank you again for your thoughtful suggestions to improve our work.
>
> We provide additional experimental results and explanations in the authors' response and revision. As the deadline for open discussion is soon, we would sincerely hope to use this opportunity to see if our responses are sufficient and if any concern remains. It will be our great pleasure if you would consider updating your review or score.
>
> Thanks again for your time.
>
> Best,
>
> Authors

---

> > ### Comment · Reviewer_Uzai · 2022-12-08
> > **Response to authors**
> >
> > Thank you for the detailed response. I have read over it and think that it should address my previous concerns.
> >
> > The paper investigates a reasonable method for trimming down model complexity along the training process, although not very surprising but did nothing wrong and indeed a solid execution.
> >
> > As committed, I would like to raise the score to weak accept.
> >
> > Best,

---

> > > ### Author Response · Authors · 2022-12-08
> > > **Author Response to Reviewer Uzai**
> > >
> > > Thank you for raising the score. Your comments are very constructive, e.g., clarifications on computation overhead and adding results for more structures such as ViT, which helps to strengthen our paper. We have addressed all your comments in our revision. Thank you again for your valuable time.
> > >
> > > Best,
> > >
> > > Authors

---

### Official Review · Reviewer_VTwh · 2022-10-31

**Confidence:** 4
**Correctness:** 3
**Technical Novelty And Significance:** 3
**Empirical Novelty And Significance:** 3
**Recommendation:** 8

**Clarity, Quality, Novelty And Reproducibility:**

The writing, while clear, is missing several important factors like reporting hyperparameters for baselines and experiment setups. In terms of novelty, the idea is interesting and novel, and in fact reveals that weight histories may be transferable and predictive of performance. For reproducibility however, the paper currently lacks several important details (see weaknesses above.),

**Strength And Weaknesses:**

**Strengths**
1. The meta-predictor is cleverly designed to consider to be layer agnostic by subsampling weights to a fixed vector.
2. The performance of SmartFrz is better than Linear or Autofreeze for the given experiments.
3. The authors perform a fairly complrehensive series of experiments showing the effect of sequence lengths.

**Weaknesses and Qeustions**
1. The main advantage of freezing sequentially is that gradients do not need to be calculated. How are the authors showing lower tFlops given that the gradients still need to be calculated for the frozen layers? I am curious to see if the implementation to see how exactly the savings occur.
2. The authors also do not mention the hyperparameters and the experimental setup for Linear and Autofreeze. This is extremely important as Autofreeze appears to be very susceptible to hyperaprameter values.
3. it appears that the attention based predictor is a Resnet50 trained on Imagenet. How transferable is this predictor to other datasets? While Cifar-10/100 are a good baseline, it would be a valuable addition to include results on a variety of datasets to confirm the performance.
4. Further, the authors claim that they use CKA to train the predictor. Do they use CKA with the subsampled weights or the original size? Also are there any thresholds that are used to generate the training dataset for the predictor? Fig. 4 shows that CKA is not a monotonic curve, and the choice of this threshold may affect the predictor adversely.
5. The authors also claim that they report the average results over 5 runs. However, there are no error bars or std deviation reported for any of the numbers. I suggest that the authors add these in order to determine the significance.
6. Weight histories are also dependent on the optimizer used.  What optimizers does SmartFrz work with?

**Summary Of The Paper:**

The paper proposes an interesting approach to predicting freezing layers during training by training a meta-predictor. The attention based meta-predictor takes in weight history and predicts if a layer should be frozen or not. The predictor is also pretrained on a dataset generated using the CKA similarities between a well trained and novel network. The authors show improvements over other layer freezing approaches that rely on heuristics for image classification on a variety of architectures.

**Summary Of The Review:**

Overall, the idea of using a meta predictor for layer freezing is inspired. However, the experiments are not documented well, and several details are missing. In addition, it is unclear why SmartFrz outperforms other freezing algorithms given that layers are still being frozen out of order. It would be also be of independent interest to consider more tasks such as language modelling and architectures like ViTs to see if the predictor is transferable.

I am currently leaning towards a weak reject, but am open to changing my opinion if the authors can provide reasonable explanations for my concerns listed above.

---

> ### Author Response · Authors · 2022-11-18
> **Author Response to Reviewer VTwh (Part 5/5)**
>
> ### **Q4. Use CKA to train the predictor? Use the subsampled weights or the original size? Any thresholds that are used to generate the training dataset? The impact of threshold value on predictor.**
>
> Thank you for the question.
>
> * **Original size and subsampled (tailored) weights**
>
> **The direct answer to the question is that the CKA is calculated based on the output feature maps from the layers with the original size. The training data in the generated dataset is a sequence of historical subsampled weights.**
>
> We would like to kindly clarify that we do NOT directly use the CKA to train the predictor.
>
> The CKA is used as the indicator for the layer similarity and is used to generate the ground-truth label of each training data (i.e., whether a given layer is stabilized or not) in the generated dataset. And the CKA is calculated based on the layer’s output feature maps, NOT the weights. The output feature map of a layer has to be obtained by conducting a layer forward propagation with the **original size**.
>
> In our generated dataset, each training sample is a sequence of historical **subsampled** weights from ResNet50, and the sequence size equals the attention window (details mentioned in Section 3.4).
>
> The reason that we use **subsampled** weights is that we intend to use only one predictor to predict all layers with different sizes. Therefore, we need to tailor the weights from different layers to the same size (i.e., subsampling weights), as we discussed in Section 3.2 in the layer tailoring part.
>
> Using subsampled weights to train the predictor will not degrade the performance of the predictor as long as using a large enough size of the subsampled weights (e.g., 1024).
>
> The reason is that, with the appropriate sizes, the subsampled weights can precisely reflect the training status of the whole weights, as mentioned in Section 3.2. We also make this clearer in the revised paper.
>
> We also provided a sensitivity analysis in our paper about the impact of the size of the subsampled weights (tailored layer size) on accuracy. Please also refer to Table 5.
>
> * **thresholds for dataset generation**
>
> Yes, there are some fluctuations in the CKA curve since the training process itself is noisy to some extent, and our threshold can effectively mitigate the impact of the fluctuations.
>
> We consider a layer stable when its CKA change rate is lower than 1%. In specific, if a layer's CKA is unstable, we will keep it training. The choice of threshold indeed affects the quality of the dataset (i.e., how well the provided ground truth balances the trade-off between accuracy and efficiency), affecting the predictor's accuracy. To alleviate the adverse effects, we use the Moving Average technique to leverage more CKA data to make decisions.
>
> We measure the quality of the training data with different threshold choices (i.e., from 0.5% to 2.0%). The experiment results are shown in Table R.16. To best train our predictor, we select the data produced by the threshold of 1%, which yields the best trade-offs between accuracies and computation overheads.
>
> >**Table R.16: Hyperparameter sweep at different CKA change rate thresholds to consider a layer converged. The results are obtained by fine-tuning ResNet50 on CIFAR-10 and CIFAR-100 datasets.**
> | Threshold 	| CIFAR-10 |           	| CIFAR-100 |           	|
> |---------------|----------|---------------|-----------|---------------|
> |           	| Accuracy | Time (Second) | Accuracy  | Time (Second) |
> | Full Training | 96.10%   | 2,594     	| 81.68%	| 2,500     	|
> | 0.5%      	| 96.17%   | 2,234     	| 81.69%	| 1,992     	|
> | 1.0%      	| 96.12%   | 1,955     	| 81.73%	| 1,787     	|
> | 1.5%      	| 95.78%   | 1,823     	| 81.22%	| 1,717     	|
> | 2.0%      	| 95.41%   | 1,755     	| 80.65%	| 1,654     	|
>
> ---
>
> ### **Q5. What optimizers does SmartFRZ work with?**
>
> We use the SGD optimizer with momentum for both the predictor and the target networks in our experiments. We make this clearer in the revised paper.
>
>
> ---
>
> ### **Q6. The standard deviation for multiple runs.**
>
> Thanks for your suggestion. We update the results in our revised paper and add the standard deviation for accuracy. Please refer to Tables 2 & 3 in the main paper and Tables 6 & 7 in the appendix.

---

> ### Author Response · Authors · 2022-11-18
> **Author Response to Reviewer VTwh (Part 4/5)**
>
> ### **Q3. The attention-based predictor is a Resnet50 trained on Imagenet. The transferable to other datasets. (cont.)**
>
> * **BERT for NLP Tasks**
>
> We evaluate our method on the NLP domain using the BERT-base model. The results are provided in Appendix A.1 and Table 6 (Page 13) in the revised paper.
>
> We also put the results in Table R.14 here.
>
> >**Table R.14: Comparison of different freezing methods in the NLP domain. The results are obtained by fine-tuning (5 epochs) BERT-base on two datasets, MRPC and CoLA.**
> | Method                              |    	MRPC    	|          	 |        CoLA    	|           	|
> |-------------------------------------|:------------------:|---------------|:------------------:|---------------|
> |                                 	| Accuracy       	| Time (Second) | Accuracy       	| Time (Second) |
> | Full Training                   	| 86.51\%$\pm$0.24\% | 316       	| 56.65\%$\pm$0.18\% | 723       	|
> | Linear Freezing                 	| 85.80\%$\pm$0.21\% | 221       	| 55.90\%$\pm$0.29\% | 429       	|
> | AutoFreeze                      	| 86.02\%$\pm$0.36\% | 266       	| 56.05\%$\pm$0.21\% | 431       	|
> | **SmartFRZ (Ours)** | **86.76\%$\pm$0.25\%** | **215**       	| **56.66\%$\pm$0.22\%** | **423**       	|
>
> As shown in Table R.14, SmartFrz is able to significantly reduce fine-tuning time without accuracy loss. Moreover, SmartFrz delivers an average of 0.86% higher accuracy than linear freezing with similar training times. Compared to AutoFreeze, SmartFrz provides 0.68% higher accuracy while consuming 10.5% less time on average. These results demonstrate the superiority of SmartFrz over linear freezing and AutoFreeze.
>
> * **Training ResNet50 from scratch on ImageNet**
>
> We also add the experiments of training ResNet50 from scratch on the ImageNet dataset. The results are provided in Table 3 in Section 4.2 (Page 8) in the revised paper.
>
> We also put the experimental results in Table R.15 here.
>
> >**Table R.15: Comparison of different freezing methods in training ResNet50 from scratch on the ImageNet dataset (100 epochs).**
> | Method      	| 	Accuracy 	|    Time (Minute)       	|
> |-----------------|:----------------:|---------------|
> | Full Training   | 76.89\%$\pm$0.12\% | 780       	|
> | Linear Freezing | 75.45\%$\pm$0.36\% | 626       	|
> | AutoFreeze  	| 74.06\%$\pm$0.31\% | 682       	|
> | **SmartFRZ (Ours)** | **76.80\%$\pm$0.26\%** | **621**     	|
>
> And we can observe from the table that SmartFRZ delivers significantly higher accuracy, 1.35% higher than linear freezing and 2.74% higher than AutoFreeze. SmartFRZ also consumes 8.9% less time than AutoFreeze.

---

> ### Author Response · Authors · 2022-11-18
> **Author Response to Reviewer VTwh (Part 3/5)**
>
> ### **Q3. The attention based predictor is a Resnet50 trained on Imagenet. The transferable to other datasets.**
>
> Thank you for your valuable question and suggestion.
>
> First, we would like to kindly clarify that we train the predictor with a generated dataset that uses ResNet50 and ImageNet. The predictor itself is a separate lightweight attention model trained offline based on the generated dataset (details described in Section 3.4).
>
> The goal of the predictor is to learn the generic convergence pattern along the training history, thanks to the attention mechanism. This is the reason that our predictor can be generalized and applied across different models and datasets.
>
> As suggested by the reviewer, to further validate the effectiveness and generalizability of our SmartFRZ, we add more results, including 1) both training from scratch and fine-tuning for **vision transformer model (DeiT)**, 2) fine-tuning **BERT model for NLP tasks**, and 3) training from scratch on **ImageNet using ResNet50**.
>
> * **Vision transformer model (DeiT-T)**
>
> We evaluate our method on DeiT-T for both fine-tuning and training from scratch scenarios. The results of fine-tuning experiments are added to Table 2 in Section 4.2 (Page 8) in our revised paper. The results of training from scratch experiments are provided in Appendix A.2 and Table 7 (Page 13) in the revised paper.
>
> We also put here the results mentioned above, as shown in Table R.12 and Table R.13.
>
> >**Table R.12: Comparison of different freezing methods in fine-tuning a vision transformer model DeiT-T. It is pre-trained on the ImageNet dataset and then fine-tuned for 100 epochs to converge.**
> | Method      	|      CIFAR-10  	|      	|         	|  	CIFAR-100 	|          |         	|
> |-----------------|:------------------:|----------|-------------|:------------------:|----------|-------------|
> |             	| Accuracy       	| Time 	| Computation | Accuracy       	| Time 	| Computation |
> |             	|                    | (Second) | (TFLOPs)	|                	| (Second) | (TFLOPs)	|
> | Full Training   | 97.48\%$\pm$0.20\% | 14,603   | 32,400  	| 85.03\%$\pm$0.28\% | 14,628   | 32,400  	|
> | Linear Freezing | 97.06\%$\pm$0.23\% | 8,760	| 16,290  	| 83.89\%$\pm$0.19\% | 9,956	| 17,542  	|
> | AutoFreeze  	| 97.35\%$\pm$0.46\% | 10,368   | 17,786      | 84.59\%$\pm$0.37\% | 11,154   | 18,710  	|
> | **SmartFRZ (Ours)** | **97.65\%$\pm$0.36\%** | **8,662**    | **15,529**      | **84.82\%$\pm$0.25\%** | **9,599**	| **16,636**  	|
>
> >**Table R.13: Comparison of different freezing methods in training DeiT-T from scratch on ImageNet dataset (300 epochs).**
> | Method                          	| Accuracy       	| Time (Minute) |
> |-------------------------------------|--------------------|---------------|
> | Full Training                       | 72.21\%$\pm$0.12\% | 2,190     	|
> | Linear Freezing                 	| 71.19\%$\pm$0.24\% | 1,817     	|
> | AutoFreeze                      	| 71.55\%$\pm$0.48\% | 1,815     	|
> | **SmartFRZ (Ours)** | **72.06\%$\pm$0.29\%** | **1,801**     	|
>
> We can observe from Table R.12 and R.13 that our SmartFRZ framework consistently outperforms linear freezing and AutoFreeze.
>
> Table R.12 shows the results of fine-tuning the vision transformer model DeiT-T. Compared to linear freezing, it delivers 0.72% higher accuracy on average while consuming slightly less training time. Compared to AutoFreeze, it delivers 0.27% higher accuracy on average with 15.2% less training time.
>
> Table R.13 shows the results of training the vision transformer model DeiT-T from scratch on the ImageNet dataset. As one can observe, SmartFRZ delivers higher accuracy (0.69\% on average) than linear freezing and AutoFreeze with the lowest training time.

---

> ### Author Response · Authors · 2022-11-18
> **Author Response to Reviewer VTwh (Part 2/5)**
>
> ### **Q2. The hyperparameters and the experimental setup for Linear and Autofreeze**
>
> Thank you for this valuable comment.
>
> For both the linear freezing and AutoFreeze baselines, we follow the recommended guideline for the hyperparameter setting in the original paper and make a hyperparameter sweep for the linear freezing and AutoFreeze. For the comparison results in our paper, we have already chosen the best hyperparameter configuration from the sweep for both linear freezing and AutoFreeze.
>
> We add detailed descriptions of the hyperparameter choices and the experimental setup for linear freezing and AutoFreeze in Appendix B (Pages 13-14) in the revised paper.
>
> Here are detailed explanations of our hyperparameter settings.
>
> **Linear freezing** employs a layer-wise cosine annealing learning rate schedule without restarts, where the first layer’s learning rate is reduced to zero partway through training (at $t_0$), and each subsequent layer’s learning rate is annealed to zero thereafter. Once a layer’s learning rate reaches zero, it is frozen and will not be updated afterward. The learning rate $\alpha_i$ of layer $l_i$ (whose learning rate anneals to zero at $t_i$) at iteration $t$ is calculated as:
>
> $\alpha_i(t) = 0.5*\alpha_i(0)(1+\cos{\pi t/t_i})$
>
> There is one hyperparameter in linear freezing, $t_0$, which defines which epoch the learning of the first layer reaches zero. The value of $t_i$ is linearly spaced between $t_0$ and the total number of epochs.
>
> We explore the impact of the value of $t_0$. Table R.10 shows the results with different configurations of $t_0$. Following the setting in the original paper, we vary the number of total training epochs for each configuration to make the total training time of each configuration almost the same. As shown in the table, with similar training time, setting $t_0$ to 0.5 provides the highest accuracy, which is also recommended by the original paper. To this end, in our main paper, we set $t_0$ to 0.5 to get the best results of linear freezing.
>
> >**Table R.10:  Hyperparameter sweep at different $t_0$ values in linear freezing. The results are obtained by fine-tuning ResNet50 on the CIFAR-10 dataset.**
> |          	                  | Accuracy | Time (Second) |
> |--------------------------------|----------|---------------|
> | $t_0$ = 0.3                	| 95.62\%  | 1,989     	|
> | $t_0$ = 0.4                	| 95.81\%  | 1,986     	|
> | $t_0$ = 0.5		 | 96.05\%  | 1,980     	|
> | $t_0$ = 0.6                	| 95.97\%  | 2,015     	|
>
> **AutoFreeze** periodically freezes layers whose gradients norm change rate is low.
>
> There are two hyperparameters in the AutoFreeze framework. The first one is the number of evaluation intervals $M$, which indicates the frequency of freezing layers. For example, 4 intervals/epoch means that AutoFreeze evaluates the gradient norm change rate for each active layer and freezes the layers with a low gradient norm change rate 4 times per epoch.
>
> The second hyperparameter is the threshold $N$ to determine whether a layer is converged or not. If a layer's gradient norm change rate is in the bottom $N^{th}$ percentile of all the active layers, it is considered converged and ready to be frozen (there is another requirement that a layer can be frozen until all the layers in front of it have been frozen).
>
> In our experiments, we follow the recommendation from the original AutoFreeze paper and set the $N$ to 50%. For a fair comparison, we further study how the number of evaluation intervals per epoch (i.e., $M$) affects the accuracy and training time.
>
> We conduct the experiments by varying the intervals per epoch from 2 to 5 since the AutoFreeze paper finds the trade-off between accuracy and training time is balanced for a range of values (2 to 5 intervals/epoch). Table R.11 shows the experimental results, and it can be observed from the table that setting the $M$ to 4 can maintain the accuracy with a high training time reduction. To this end, in our main paper, we set the $N$ to 50% and $M$ to 4 to get the best results of AutoFreeze.
>
> >**Table R.11: Hyperparameter sweep at different numbers of evaluation intervals per epoch in AutoFreeze. The results are obtained by fine-tuning ResNet50 on the CIFAR-10 dataset.**
> |     	| Accuracy | Time (Second) |
> |---------|----------|---------------|
> | $M$ = 2 | 96.19\%  | 2,536     	|
> | $M$ = 3 | 96.28\%  | 2,487     	|
> | $M$ = 4 | 96.24\%  | 2,424     	|
> | $M$ = 5 | 95.69\%  | 2,378     	|

---

> ### Author Response · Authors · 2022-11-18
> **Author Response to Reviewer VTwh (Part 1/5)**
>
> **We appreciate the valuable comments from the reviewer. We appreciate the reviewer's acknowledgment that our proposed ideas are interesting and novel. We carefully address all the reviewer’s questions and revise the paper accordingly. We hope our response can help alleviate the reviewer's concern.**
>
> ---
>
> ### **Q1. How to achieve lower TFLOPs with non-sequential freezing? How to implement it?**
>
> Here is a brief summary of the answer.
>
> **Lower TFLOPS:**
>
> 1. Freezing a layer whose earlier layers have not been frozen can still save the computation of the **gradient of weights**.
>
> 2. Layers are not converged sequentially during training. Using non-sequential freezing can freeze the layers as long as they are stable, and does not need to wait until all its earlier layers are frozen.
>
> These two reasons bring the superiority of our SmartFRZ in FLOPs reduction, acceleration, and high accuracy.
>
> **Actual Implementation:**
>
> We use PyTorch. To freeze a layer, we only need to set the attribute **requires_grad** of this layer to **False**. PyTorch will automatically update the computational graph and eliminate unnecessary computation. It can be easily incorporated into different training codes, making our method an easy-to-use tool in practice.
>
> We provide a detailed explanation below:
>
> As stated in Section 2 (Page 3) in the paper, the backward propagation consists of two parts of computations: i) calculating the **gradients of activations** and ii) calculating the **gradients of weights**.
>
> **In sequential freezing**, both the computation of gradients of activations and gradients of weights can be avoided for all the frozen layers.
>
> **In non-sequential freezing**, suppose some layers earlier than a frozen layer are still actively trained; we still need to calculate the gradients of activations of this frozen layer to maintain the back-propagation, but we can still eliminate the computation cost of the gradients of weights since the weights in the layer will not be updated anymore, and it will not affect the back-propagation. Therefore, it is worth freezing a stabilized layer as early as possible and does not need to wait until all its earlier layers are frozen.
>
> **For the implementation**, we implement our SmartFRZ framework using PyTorch. With PyTorch, if we want to freeze a converged layer, we only need to set the attribute “requires_grad” of all parameters of this layer to False. The PyTorch will automatically update the computational graph and eliminate unnecessary computation.
>
> Here is a pseudo-code snippet:
>
> ~~~~
> for params in layer_to_freeze.parameters():
>       params.requires_grad = False
> ~~~~
>
> Our SmartFRZ can be easily incorporated into different training codes, making our SmartFRZ an easy-to-use tool in practice.

---

> ### Author Response · Authors · 2022-12-01
> **Author Response to Reviewer VTwh**
>
> Dear Reviewer VTwh,
>
> Thanks for your time and reviewing efforts! We appreciate your constructive comments.
>
> We provide suggested results in the authors' response, including the experiments on more models and datasets to show the generalizability, details of hyperparameters selection and experimental setup of Linear Freezing and AutoFreeze approaches that we compared, std deviation of our experimental results, and clarification of the computation reduction and how we utilize CKA.
>
> We hope our responses have answered your questions. It would be our great pleasure if you would consider updating your review or score.
>
> Best,
>
> Authors

---

> > ### Comment · Reviewer_VTwh · 2022-12-04
> > **Response**
> >
> > I thank the authors for responding to my concerns and adding additional studies. The additional experiments do suggest that the meta-predictor has learned transferable weight trajectory properties, which is a very interesting insight.
> >
> > The authors have also addressed most of my concerns and I am therefore raising my score.

---

> > > ### Author Response · Authors · 2022-12-04
> > > **Author Response**
> > >
> > > Thank you for raising the score.
> > > Your comments are very constructive, e.g., clarifications of the baseline's experimental setup and adding results for more networks and datasets, which make our paper stronger. We have addressed all your comments in our revision.
> > > Thank you again for your valuable time.

---

### Official Review · Reviewer_D3g6 · 2022-10-31

**Confidence:** 4
**Correctness:** 3
**Technical Novelty And Significance:** 3
**Empirical Novelty And Significance:** 2
**Recommendation:** 6

**Clarity, Quality, Novelty And Reproducibility:**

I list the questions and concerns below:

1. According to Figure 1, the attention-based predictor must access all previous timestamps' previous weights (historical weights). If $t$ is large, the SmartFRZ algorithm may struggle with the memory problem, unlike Linear Freezing and AutoFreeze.

2. Can the author provide more detailed motivations for using attention in the predictor architecture? Why would an attention module be a better choice than regular LSTM-based architecture? It is well-known attention module is notorious for quadratic complexity.

3. In Section 3.2 Layer Taioloring paragraph, I am not aware Lottery Ticket Hypothesis (LTH) paper mentions "the over-parameterization not only exists in the inference process but also in the training process." The hypothesis says there exists an initialization that the sub-network matches the original network after, at most, training the same number of iterations. To my understanding, LTH is an entirely different statement from what the author is trying to convey. Furthermore, I do not buy the line "the gradient distribution is one of the indicators that best characterizes the features of parameters during the training process." Can the author refer to the literature supporting this claim?

4. The biggest weakness of the paper is testing their algorithms on a simple dataset, such as CIFAR. The motivation for freezing business is to reduce the overhead of training procedure, which the CIFAR dataset usually do not need. We cannot extrapolate the SmartFRZ empirical results collected on only the CIFAR and CNN architecture to the different classes of architectures and other tasks. In order to empirically show that SmartFRZ is better than AutoFreeze, can the author also make comparisons on NLP tasks listed in the AutoFreeze paper?

**Strength And Weaknesses:**

Strengths
1. The author shows that SmartFRZ achieves performance more efficient training framework than existing algorithms, such as Linear Freezing and AutoFreeze.
2. The paper is well written in general.

Weakness
1. SmartFRZ needs to train the predictor separately from the actual network training.
2. Limited comparisons only on CIFAR and two CNN architectures.
3. Lack of explanations on the motivation for the suggested approaches.


**Summary Of The Paper:**

This paper proposes a layer freezing method to improve training efficiency by introducing attention-guided layers. SmartFRZ aggregates the historical weights using the attention-based predictor to allow the dynamic decision of freezing layers leading to the efficient training framework. The authors support their methods through empirical comparisons of existing algorithms.

**Summary Of The Review:**

The authors provide empirical results regarding the efficient training framework on neural networks. However, their motivations for the methodology are not convincing to me, and their results are limited to a specific dataset (CIFAR) and the family of convolutional neural networks. Furthermore, the author's explanation of their approach needs to be clearer. Overall, it is a well-written paper, but I do not think the paper does not meet the ICLR bars.

---

> ### Author Response · Authors · 2022-11-18
> **Author Response to Reviewer D3g6 (Part 4/4)**
>
> ### **Q5. Unclear about the statement of the “lottery ticket hypothesis” and the claim of “the gradient distribution …”.**
>
> Sorry for the confusion.
>
> For the layer tailoring part in Section 3.2, we try to solve the potential issue of the extra memory costs introduced by the predictor. Instead of assigning individual predictors for each layer, we want to use only one predictor to serve different layers that have different weight sizes.
>
> **For the statement of LTH**, we intend to convey that we are inspired by recent successes in the field of sparse training [3][4], which demonstrates that not all the weights in a layer are indispensable in training to achieve good accuracy.
> Similarly, in our case, it is possible only to use partial weights of a layer to represent the layer's learning status.
>
> Therefore, we conduct an experiment to observe the relationship between a layer's gradient subsets and all its gradients.
>
> **For the claim that** “the gradient distribution is one of the indicators that best characterizes the features of parameters during the training process.”
> We intend to say that we observe the gradient distribution because the gradients are the most direct factor determining the weight update status in the training process.
>
> And we observe that, for a network layer, the gradient distribution of the randomly selected weights subset is highly similar to the gradient distribution of all the weights.
> This characteristic provides us with the feasibility to design the layer tailoring technique.
> With our layer tailoring, the predictor can be shared by the layers with different sizes. And the results demonstrate the effectiveness of our layer tailoring technique.
>
> We have carefully revised the paper to make it clearer. Please refer to the Section 3.2 layer tailoring part in the revised paper.
>
> [3] Frankle, Jonathan, et.al., “The lottery ticket hypothesis: Finding sparse, trainable neural networks.” ICLR 2019.
>
> [4] Utku Evci, et.al., “Rigging the Lottery: Making All Tickets Winners”, ICML 2020.

---

> > ### Comment · Reviewer_D3g6 · 2022-11-22
> > **Response**
> >
> > Thank you for the response. The authors have provided additional experiments to resolve my concern. Therefore, I raise the score.

---

> > > ### Author Response · Authors · 2022-11-22
> > > **Author Response**
> > >
> > > Thank you for raising the score. Your comments are very constructive, e.g., adding results for more networks and datasets, and adding explanations on the motivation,  which make our paper stronger. We have addressed all your comments in our revision. And thank you again for your valuable time.

---

> ### Author Response · Authors · 2022-11-18
> **Author Response to Reviewer D3g6 (Part 3/4)**
>
> ### **Q3. More results for other models and tasks. (cont.)**
>
> * **Training ResNet50 from scratch on ImageNet**
>
> We also add the experiments of training ResNet50 from scratch on the ImageNet dataset. The results are provided in Table 3 in Section 4.2 (Page 8) in the revised paper.
>
> We also put the experimental results in Table R.8 here.
>
> >**Table R.8: Comparison of different freezing methods in training ResNet50 from scratch on ImageNet dataset (100 epochs).**
> | Method      	| 	Accuracy 	|   Time (Minute)        	|
> |-----------------|:----------------:|---------------|
> | Full Training   | 76.89\%$\pm$0.12\% | 780       	|
> | Linear Freezing | 75.45\%$\pm$0.36\% | 626       	|
> | AutoFreeze  	| 74.06\%$\pm$0.31\% | 682       	|
> | **SmartFRZ (Ours)** | **76.80\%$\pm$0.26\%** | **621**     	|
>
> And we can observe from the table that SmartFRZ delivers significantly higher accuracy, with 1.35% higher than linear freezing and 2.74% higher than AutoFreeze. SmartFRZ also consumes 8.9% less time than AutoFreeze.
>
>
> ---
>
> ### **Q4. Lack of explanations on the motivation for the suggested approaches.**
>
> We use the attention-based predictor because the training model does not converge to the optimal solution monotonously and consequently introduces noisy weight history information. (We discussed this in Section 3.2, and show an example in Section 3.3, Figure 4). So the predictor needs to focus on the timestamps containing more critical information and pay less attention to those with noisy/less-optimal weight histories.
>
> We use attention instead of LSTM due to two reasons: i) Compared to the LSTM-based sequence classifier, which encodes the whole input sequence into a fixed-length vector at the latest time stamp, the attention-based classifier can encode the input sequence into a sequence of vectors and directly gather information from important timestamps adaptively [1]. This enhances the adaptivity and generalizability of our predictor. ii) Compared to the LSTM-based sequence classifier, which encodes timestamps one-by-one and introduces long-dependency and time cost, the attention-based classifier can process timestamps in parallel, thus leading to less time overhead [2].
>
> We also compare our attention-based predictor with an LSTM-based predictor, as shown in Table R.9 Experimental results show that our attention-based approach leads to 0.72% higher accuracy with less training time.
>
> >**Table R.9: Comparison of attention-based predictor and LSTM-based predictor.**
> | Model	| Predictor   	| CIFAR-10 |           	| CIFAR-100 |           	|
> |----------|-----------------|----------|---------------|-----------|---------------|
> |      	|             	| Accuracy | Time (Second) | Accuracy  | Time (Second) |
> | ResNet50 | LSTM-based  	| 95.52%   | 1,998     	| 81.67%	| 1,765     	|
> |      	| Attention-based | 96.12%   | 1,955     	| 81.73%	| 1,787    	 |
> | VGG11	| LSTM-based  	| 92.31%   | 1,587     	| 73.70%	| 1,862     	|
> |      	| Attention-based | 93.58%   | 1,554     	| 74.66%	| 1,831     	|
>
> In terms of complexity, while the attention mechanism usually causes quadratic complexity in the sequence-to-sequence tasks, our predictor handles sequence classification (i.e., sequence-to-one) tasks instead of the sequence-to-sequence task, which gives linear complexity. The encoding stage (i.e., Equation 1) encodes each time stamp, yielding $O(t)$ complexity. The aggregating stage (i.e., Equation 2&3) calculates weights from the current timestamp to sequences of timestamps as a one-to-all procedure, and it causes $O(t)$ as well. Furthermore, the last MLP classification steps only cause $O(1)$. In summary, our predictor has linear complexity ($\sim$$O(t)$) in terms of sequence length instead of quadratic complexity ($\sim$$O(t^2)$).
>
> [1] Bahdanau, Dzmitry, et.al., "Neural machine translation by jointly learning to align and translate." arXiv preprint arXiv:1409.0473 (2014).
>
> [2] Vaswani, Ashish, et al., "Attention is all you need." Advances in neural information processing systems 30 (2017).

---

> ### Author Response · Authors · 2022-11-18
> **Author Response to Reviewer D3g6 (Part 2/4)**
>
> ### **Q3. More results for other models and tasks.**
>
> Thank you for the valuable feedback.
>
> To further validate the effectiveness of our SmartFRZ, we add more results, including **the BERT model for NLP, the vision transformer model (DeiT)**, and **training from scratch on ImageNet using ResNet50**.
>
> * **BERT for NLP Tasks**
>
> We evaluate our method on the NLP domain using the BERT-base model. The results are provided in Appendix A.1 and Table 6 (Page 13) in the revised paper.
>
> We also put the results in Table R.5 here.
>
> >**Table R.5: Comparison of different freezing methods in the NLP domain. The results are obtained by fine-tuning (5 epochs) BERT-base on two datasets, MRPC and CoLA.**
> | Method                              |    	MRPC    	|          	 |        CoLA    	|           	|
> |-------------------------------------|:------------------:|---------------|:------------------:|---------------|
> |                                 	| Accuracy       	| Time (Second) | Accuracy       	| Time (Second) |
> | Full Training                   	| 86.51\%$\pm$0.24\% | 316       	| 56.65\%$\pm$0.18\% | 723       	|
> | Linear Freezing                 	| 85.80\%$\pm$0.21\% | 221       	| 55.90\%$\pm$0.29\% | 429       	|
> | AutoFreeze                      	| 86.02\%$\pm$0.36\% | 266       	| 56.05\%$\pm$0.21\% | 431       	|
> | **SmartFRZ (Ours)** | **86.76\%$\pm$0.25\%** | **215**       	| **56.66\%$\pm$0.22\%** | **423**       	|
>
> As shown in Table R.5, SmartFrz is able to significantly reduce fine-tuning time without accuracy loss. Moreover, SmartFrz delivers an average of 0.86% higher accuracy than linear freezing with similar training times. Compared to AutoFreeze, SmartFrz provides 0.68% higher accuracy while consuming 10.5% less time on average. These results demonstrate the superiority of SmartFrz over linear freezing and AutoFreeze.
>
> * **Vision transformer model (DeiT-T)**
>
> We evaluate our method on DeiT-T for both fine-tuning and training from scratch scenarios. The results of fine-tuning experiments are added to Table 2 in Section 4.2 (Page 8) in our revised paper. The results of training from scratch experiments are provided in Appendix A.2 and Table 7 (Page 13) in the revised paper.
>
> We also put the results mentioned above, as shown in Table R.6 and Table R.7.
>
> >**Table R.6: Comparison of different freezing methods in fine-tuning a vision transformer model DeiT-T. It is pre-trained on the ImageNet dataset and then fine-tuned for 100 epochs to converge.**
> | Method      	|      CIFAR-10  	|      	|         	|  	CIFAR-100 	|          |         	|
> |-----------------|:------------------:|----------|-------------|:------------------:|----------|-------------|
> |             	| Accuracy       	| Time 	| Computation | Accuracy       	| Time 	| Computation |
> |             	|                    | (Second) | (TFLOPs)	|                	| (Second) | (TFLOPs)	|
> | Full Training   | 97.48\%$\pm$0.20\% | 14,603   | 32,400  	| 85.03\%$\pm$0.28\% | 14,628   | 32,400  	|
> | Linear Freezing | 97.06\%$\pm$0.23\% | 8,760	| 16,290  	| 83.89\%$\pm$0.19\% | 9,956	| 17,542  	|
> | AutoFreeze  	| 97.35\%$\pm$0.46\% | 10,368   | 17,786      | 84.59\%$\pm$0.37\% | 11,154   | 18,710  	|
> | **SmartFRZ (Ours)** | **97.65\%$\pm$0.36\%** | **8,662**    | **15,529**      | **84.82\%$\pm$0.25\%** | **9,599**	| **16,636**  	|
>
> >**Table R.7: Comparison of different freezing methods in training DeiT-T from scratch on ImageNet dataset (300 epochs).**
> | Method                          	| Accuracy       	| Time (Minute) |
> |-------------------------------------|--------------------|---------------|
> | Full Training                       | 72.21\%$\pm$0.12\% | 2,190     	|
> | Linear Freezing                 	| 71.19\%$\pm$0.24\% | 1,817     	|
> | AutoFreeze                      	| 71.55\%$\pm$0.48\% | 1,815     	|
> | **SmartFRZ (Ours)** | **72.06\%$\pm$0.29\%** | **1,801**     	|
>
> We can observe from Tables R.6 and R.7 that our SmartFRZ framework consistently outperforms linear freezing and AutoFreeze.
>
> Table R.6 shows the results of fine-tuning the vision transformer model DeiT-T. Compared to linear freezing, it delivers 0.72% higher accuracy on average while consuming slightly less training time. Compared to AutoFreeze, it delivers 0.27% higher accuracy on average with 15.2% less training time.
>
> Table R.7 shows the results of training the vision transformer model DeiT-T from scratch on the ImageNet dataset. As one can observe, SmartFRZ delivers higher accuracy (0.69\% on average) than linear freezing and AutoFreeze with the lowest training time.

---

> ### Author Response · Authors · 2022-11-18
> **Author Response to Reviewer D3g6 (Part 1/4)**
>
> **We appreciate the valuable comments from the reviewer. We carefully address all the reviewer’s questions and revise the paper accordingly. We hope our response can help alleviate the reviewer's concern.**
>
> ---
>
>
> ### **Q1. Needs to train the predictor separately.**
>
> The predictor is trained offline once to learn the generic convergence pattern. The predictor training only needs to use a single generated dataset (we only use ImageNet + ResNet50 to generate it). The trained predictor can be **generalized across different models and datasets without fine-tuning**. When using the trained predictor for guiding the layer freezing during the actual network training process, it does not require extra computation costs to train the predictor.
>
> ---
>
> ### **Q2. The concern of memory costs by accessing previous timestamps' previous weights.**
>
> Thank you for this valuable question.
>
> We are aware of the potential challenge of the extra memory costs introduced by the attention-based predictor. To avoid this issue, our proposed predictor is a lightweight design, and the proposed layer tailoring technique can effectively make the memory overhead negligible. Instead of storing all the weights in each layer, we tailor the layer by subsampling the layer’s weights to a fixed-size vector (e.g., a size of 1024) to represent the whole layer.
>
> We added a sensitivity study vector length is shown in Table 5 in Section 4.3 (Page 9 in the revised paper). In this way, it will cost 4 KB to record one layer for one timestamp. We also set up an attention window to control timestamp length instead of all previous timestamps (e.g., 30). The sensitivity study on attention window size is shown in Table 4 in Section 4.3 (Page 9 in the revised paper). In this case, taking ResNet50 as an example, it takes only about **6 MB** to store the historical weights, which can be covered by the memory saved by layer freezing (shown in Figure 5a in the revised paper).

---

### Official Review · Reviewer_Spyk · 2022-11-01

**Confidence:** 4
**Correctness:** 4
**Technical Novelty And Significance:** 3
**Empirical Novelty And Significance:** 3
**Recommendation:** 8

**Clarity, Quality, Novelty And Reproducibility:**

Clarity: Good, but can be improved. Please address the questions asked above.

Quality: Excellent.

Novelty: Good idea to learn attention layers for freezing

Reproducibility: Excellent

**Strength And Weaknesses:**

* Strengths:
- Paper is well written and easy to follow
- Experiments are extensive, with sensitivity studies, comparison with state of the art, and computation overhead measurements.
- Proposed method is simple but effective

* Weaknesses:
- Analysis limited to CNN vision models. Unclear how well it will translate to other architectures (e.g. Transformers) or domains (e.g. Language) or objectives (e.g. Regression, Object detection)
- A few clarifying questions:
--- it is not clear how the layer freezing works dynamically given that all layers below the chosen layer needs to be frozen as well. This is mentioned in passing in Section 2, but not mentioned in methods.
--- It is not clear what the training and test datasets are. It is mentioned in passing that ImageNet + ResNet50 is used for training, is that all the training required?

**Summary Of The Paper:**

The paper proposes a layer freezing method to reduce computational time for training of ML models. An attention-based layer freezing model takes sampled parameters from a layer as input, and predicts if it should be frozen or not. The layers get dynamically frozen over the training period. Results should significant improvement over hand-crafted and heuristic methods.

**Summary Of The Review:**

Overall, a good paper with substantial improvement over baselines.

---

> ### Author Response · Authors · 2022-11-18
> **Author Response to Reviewer Spyk (Part 2/2)**
>
> ### **Q1. More results for other models and tasks. (cont.)**
>
> * **Training ResNet50 from scratch on ImageNet**
>
> We also add the experiments of training ResNet50 from scratch on the ImageNet dataset. The results are provided in Table 3 in Section 4.2 (Page 8) in the revised paper.
>
> We also put the experimental results in Table R.4 here.
>
> >**Table R.4: Comparison of different freezing methods in training ResNet50 from scratch on ImageNet dataset (100 epochs).**
> | Method      	| 	Accuracy 	|      Time (Minute)     	|
> |-----------------|:----------------:|---------------|
> | Full Training   | 76.89\%$\pm$0.12\% | 780       	|
> | Linear Freezing | 75.45\%$\pm$0.36\% | 626       	|
> | AutoFreeze  	| 74.06\%$\pm$0.31\% | 682       	|
> | **SmartFRZ (Ours)** | **76.80\%$\pm$0.26\%** | **621**     	|
>
> And we can observe from the table that SmartFRZ delivers significantly higher accuracy, with 1.35% higher than linear freezing and 2.74% higher than AutoFreeze. SmartFRZ also consumes 8.9% less time than AutoFreeze.
>
> ---
>
> ### **Q2. How does layer freezing work dynamically?**
>
> Unlike previous works that require the layers to be frozen in a sequential manner, our SmartFRZ framework does not require all the layers below (earlier than) the chosen layer to be frozen. We employ the attention-based predictor to predict whether a layer is converged and freeze it as long as it converges. The rationale behind this is based on our observation of the non-sequential convergence property of the network layers during the training process.
>
> As we show in Figure 4 in Section 3.4 (Page 6), it is interesting to see that the layer in the back possibly stabilizes faster than the front layer (e.g., layer 30 stabilizes faster than layers 10 and 20). This observation demonstrates the need for an adaptive layer freezing method, which is exactly what the SmartFRZ does, rather than forcing sequential freezing from front to back. Please also refer to the detailed discussion in the paper (Section 3.4).
>
> For the computation reduction perspective, as stated in Section 2 (Page 3), the backward propagation consists of two parts of computations: i) **calculating the gradients of activations** and ii) **calculating the gradients of weights**.
>
> If there exist some layers below a frozen layer still actively being trained, we still need to calculate the gradients of activations of this frozen layer to maintain the data pass backward propagation, but we can eliminate the calculation of gradients of weights since the weights in the layer will not be updated anymore. Therefore, it is always worth freezing a stable layer as early as possible, even if some layers below it are not frozen. And this is the advantage of our SmartFRZ, which the previous methods cannot appropriately capture.
>
> ---
>
> ### **Q3. It is not clear what the training and test datasets are. It is mentioned in passing that ImageNet + ResNet50 is used for training, is that all the training required?**
>
> Yes, only the ImageNet+ResNet50 is used to **generate** the dataset to train our layer freezing predictor. The predictor is trained offline once to learn the generic convergence pattern along the training history, which can be generalized across different models and datasets. In other words, we do NOT re-train the predictor when applying it to other benchmarks. And our experimental results demonstrate the effectiveness of our method.
>
> We make this clearer in the revised paper.

---

> > ### Comment · Reviewer_Spyk · 2022-11-21
> > **Concerns addressed**
> >
> > Thank you for the detailed experiments and clarifications. The responses have addressed the concerns I raised.

---

> > > ### Author Response · Authors · 2022-11-22
> > > **Thanks for the positive feedback**
> > >
> > > We would like to thank you again for your time and your positive rating. This is a great affirmation of our work.

---

> ### Author Response · Authors · 2022-11-18
> **Author Response to Reviewer Spyk (Part 1/2)**
>
> **We would like to thank the reviewer for the positive feedback and valuable review. We add more results as suggested by the reviewer, including the performance of our SmartFRZ on the vision transformer model (DeiT), the BERT model for NLP, and training from scratch on ImageNet using ResNet50. We clarify the questions and revise the paper carefully. The detailed answers to the questions are provided as follows. We hope our response can help clarify the reviewer's questions.**
>
> ---
>
> ### **Q1. More results for other models and tasks.**
>
> * **Vision transformer model (DeiT-T)**
>
> We evaluate our method on DeiT-T for both fine-tuning and training from scratch scenarios. The results of fine-tuning experiments are added to Table 2 in Section 4.2 (Page 8) in our revised paper. The results of training from scratch experiments are provided in Appendix A.2 and Table 7 (Page 13) in the revised paper.
>
> We also put here the results mentioned above, as shown in Table R.1 and Table R.2.
>
> >**Table R.1: Comparison of different freezing methods in fine-tuning a vision transformer model DeiT-T. It is pre-trained on ImageNet dataset and then fine-tuned for 100 epochs to converge.**
> | Method      	|      CIFAR-10  	|      	|         	|  	CIFAR-100 	|          |         	|
> |-----------------|:------------------:|----------|-------------|:------------------:|----------|-------------|
> |             	| Accuracy       	| Time 	| Computation | Accuracy       	| Time 	| Computation |
> |             	|                    | (Second) | (TFLOPs)	|                	| (Second) | (TFLOPs)	|
> | Full Training   | 97.48\%$\pm$0.20\% | 14,603   | 32,400  	| 85.03\%$\pm$0.28\% | 14,628   | 32,400  	|
> | Linear Freezing | 97.06\%$\pm$0.23\% | 8,760	| 16,290  	| 83.89\%$\pm$0.19\% | 9,956	| 17,542  	|
> | AutoFreeze  	| 97.35\%$\pm$0.46\% | 10,368   | 17,786      | 84.59\%$\pm$0.37\% | 11,154   | 18,710  	|
> | **SmartFRZ (Ours)** | **97.65\%$\pm$0.36\%** | **8,662**    | **15,529**      | **84.82\%$\pm$0.25\%** | **9,599**	| **16,636**  	|
>
> >**Table R.2: Comparison of different freezing methods in training DeiT-T from scratch on ImageNet dataset (300 epochs).**
> | Method                          	| Accuracy       	| Time (Minute) |
> |-------------------------------------|--------------------|---------------|
> | Full Training                       | 72.21\%$\pm$0.12\% | 2,190     	|
> | Linear Freezing                 	| 71.19\%$\pm$0.24\% | 1,817     	|
> | AutoFreeze                      	| 71.55\%$\pm$0.48\% | 1,815     	|
> | **SmartFRZ (Ours)** | **72.06\%$\pm$0.29\%** | **1,801**     	|
>
> We can observe from Table R.1 and R.2 that our SmartFRZ framework consistently outperforms linear freezing and AutoFreeze.
>
> Table R.1 shows the results of fine-tuning the vision transformer model DeiT-T. Compared to linear freezing, it delivers 0.72% higher accuracy on average while consuming slightly less training time. Compared to AutoFreeze, it delivers 0.27% higher accuracy on average with 15.2% less training time.
>
> Table R.2 shows the results of training the vision transformer model DeiT-T from scratch on the ImageNet dataset. As one can observe, SmartFRZ delivers higher accuracy (0.69\% on average) than linear freezing and AutoFreeze with the lowest training time.
>
> * **BERT for NLP Tasks**
>
> We evaluate our method on the NLP domain using the BERT-base model. The results are provided in Appendix A.1 and Table 6 (Page 13) in the revised paper.
>
> We also put the results in Table R.3 here.
>
> >**Table R.3: Comparison of different freezing methods in the NLP domain. The results are obtained by fine-tuning (5 epochs) BERT-base on two datasets, MRPC and CoLA.**
> | Method                              |    	MRPC    	|          	 |        CoLA    	|           	|
> |-------------------------------------|:------------------:|---------------|:------------------:|---------------|
> |                                 	| Accuracy       	| Time (Second) | Accuracy       	| Time (Second) |
> | Full Training                   	| 86.51\%$\pm$0.24\% | 316       	| 56.65\%$\pm$0.18\% | 723       	|
> | Linear Freezing                 	| 85.80\%$\pm$0.21\% | 221       	| 55.90\%$\pm$0.29\% | 429       	|
> | AutoFreeze                      	| 86.02\%$\pm$0.36\% | 266       	| 56.05\%$\pm$0.21\% | 431       	|
> | **SmartFRZ (Ours)** | **86.76\%$\pm$0.25\%** | **215**       	| **56.66\%$\pm$0.22\%** | **423**       	|
>
> As shown in Table R.3, SmartFrz is able to significantly reduce fine-tuning time without accuracy loss. Moreover, SmartFrz delivers an average of 0.86% higher accuracy than linear freezing with similar training times. Compared to AutoFreeze, SmartFrz provides 0.68% higher accuracy while consuming 10.5% less time on average. These results demonstrate the superiority of SmartFrz over linear freezing and AutoFreeze.

---

### Decision · Program_Chairs · 2023-01-20

**Decision:**

Accept: notable-top-25%

**Justification For Why Not Higher Score:**

See summary for a description of strengths. The idea is nice, but in hindsight, fairly intuitive and the natural "next step" in this line of work. I commend the authors for doing a good job of executing the idea, and certainly the technique should be spotlighted. I don't think there is enough to say here that merits a full-length Oral

**Justification For Why Not Lower Score:**

See summary for details; I personally believe that this paper is worthy of acceptance, and that the technique is conceptually simple enough for a large subset of ICLR attendees to be interested in using it. Therefore, I recommend a spotlight.

**Metareview: Summary, Strengths And Weaknesses:**

The paper proposes a new approach for improving neural network training efficiency via layer freezing. Previous such approaches have chosen heuristics to decide which layers to freeze; this paper instead proposes a simple "meta" network (which leverages self-attention) to predict which layers should be frozen (and when). This meta-network is lightweight and trained with randomly sampled "weight histories" along with centered kernel alignment scores which give an indication of whether the weights have stabilized.

This is a simple but effective idea (and beats existing freezing based approaches), which the authors support via experiments on a range of model architectures and datasets. Since the overhead of training this meta-predictor is quite low, the method is likely to be widely applicable.

Originally, a few reviewers raised concerns. Primarily, the original paper limited its experimental results to CNN models and image classification on CIFAR-10. However, during the discussion phase the authors provided a large number of additional results on transformer architectures, recurrent architectures, language models, other vision datasets, etc, which significantly strengthened the paper. The authors were also able to satisfactorily address other concerns related to conceptual benefits of the proposed approach.

Overall, this is a well-written paper that proposes a simple idea which works well in several settings.
Recommendation: accept.

**Note From Pc:**

if the above contains the word "oral" or "spotlight" please see: "oral" presentation means -> notable-top-5% and "spotlight" means -> notable-top-25%. As stated in our emails, we are disassociating presentation type from AC recommendations

**Summary Of Ac-Reviewer Meeting:**

At the end of Phase I discussion, the average score was barely above the borderline bucket (with a small spread); however, most reviewers were satisfied with the thorough responses by the authors, and bumped up their scores (thereby raising it to a unanimous accept). Therefore, we did not need to have an AC-reviewer meeting.